# Advances in Non-Animal Testing Approaches towards Accelerated Clinical Translation of Novel Nanotheranostic Therapeutics for Central Nervous System Disorders

**DOI:** 10.3390/nano11102632

**Published:** 2021-10-07

**Authors:** Mark J. Lynch, Oliviero L. Gobbo

**Affiliations:** School of Pharmacy and Pharmaceutical Sciences, Panoz Building, Trinity College Dublin, D02 PN40 Dublin, Ireland

**Keywords:** nanotheranostics, blood–brain barrier, advanced drug delivery, in vitro modelling, organ-on-chip, in silico testing

## Abstract

Nanotheranostics constitute a novel drug delivery system approach to improving systemic, brain-targeted delivery of diagnostic imaging agents and pharmacological moieties in one rational carrier platform. While there have been notable successes in this field, currently, the clinical translation of such delivery systems for the treatment of neurological disorders has been limited by the inadequacy of correlating in vitro and in vivo data on blood–brain barrier (BBB) permeation and biocompatibility of nanomaterials. This review aims to identify the most contemporary non-invasive approaches for BBB crossing using nanotheranostics as a novel drug delivery strategy and current non-animal-based models for assessing the safety and efficiency of such formulations. This review will also address current and future directions of select in vitro models for reducing the cumbersome and laborious mandate for testing exclusively in animals. It is hoped these non-animal-based modelling approaches will facilitate researchers in optimising promising multifunctional nanocarriers with a view to accelerating clinical testing and authorisation applications. By rational design and appropriate selection of characterised and validated models, ranging from monolayer cell cultures to organ-on-chip microfluidics, promising nanotheranostic particles with modular and rational design can be screened in high-throughput models with robust predictive power. Thus, this article serves to highlight abbreviated research and development possibilities with clinical translational relevance for developing novel nanomaterial-based neuropharmaceuticals for therapy in CNS disorders. By generating predictive data for prospective nanomedicines using validated in vitro models for supporting clinical applications in lieu of requiring extensive use of in vivo animal models that have notable limitations, it is hoped that there will be a burgeoning in the nanotherapy of CNS disorders by virtue of accelerated lead identification through screening, optimisation through rational design for brain-targeted delivery across the BBB and clinical testing and approval using fewer animals. Additionally, by using models with tissue of human origin, reproducible therapeutically relevant nanomedicine delivery and individualised therapy can be realised.

## 1. Introduction

The diagnosis and treatment of central nervous system (CNS) disorders constitute a notable challenge in the field of modern therapeutics and advanced drug delivery systems, and it would appear such disorders are on the rise despite increasing appreciation and elucidation of underlying aetiology and pathophysiological mechanisms [1,2] The CNS therapeutics market is set to grow to EUR 114.4 billion in 2025, and a resurgence in the neuroscience field is predicted, which will be bolstered by novel drug delivery systems and new chemical entities (NCE’s). The most recent global burden of disease (GBD) study published in the Lancet journal shows that the global burden of neurological disorder approaches the 14% value of overall disease modelled over a decade ago, accounting for 250,000 deaths relating to brain and central nervous system cancer and 100 million disability-adjusted life years relating to neurological disorders [3]. These figures mirror the 2016 systematic review of GBD 1990–2016 [4], which found that deaths increased by 39% and DALYs by 27% over this period, and that reductions were only seen for infectious causes (encephalitis, meningitis and tetanus). 

This considerable mortality and disease burden particularly in relation to chronic disability means that prompt and efficient intervention is required at the earliest possible stages to improve clinical outcomes and prognosis for affected patients, which is likely to have greater urgency due to the increasing median age of the worldwide population. As Eroom’s law would suggest [5], much of the empirical regimens available to clinicians constitute the vast majority of those agents that will be readily available for development and marketability, and so the pharmaceutical fraternity is tasked with turning to novel delivery systems for delivery of this suite of potent agents. The inefficiency of delivery and consequent inadequacy of conventional formulations in empirical regimens is largely due to the presence of the blood–brain barrier (BBB), and indeed this has been the culprit for many novel entities failing to reach clinical translation as they cannot bypass this robust physical barrier [6,7,8].

Nanomedicines have constituted one of the major breakthroughs in such novel drug delivery efforts, and has been the focus of intensified efforts in the past twenty years. Although they are not the “magic bullet” purported by Nobel laureate Paul Ehrlich as substances that seek out specific disease causing agents, they have led to a notable advancement particularly in the field of oncological diagnostics and chemotherapy [9]. One of their fundamental limitations is in delivery efficiency, as mean nanoparticle delivery efficiency is in the region of 0.7% to 5% for a single intravenous administered dose, which primarily relies on passive targeting approaches [10]. Targeted delivery thus requires the development of rational nanocarriers that are functionalised to actively and specifically reach the principal organ of interest following administration in therapeutically significant concentrations and, in the case of the brain, to cross the BBB.

The culmination of such efforts has arisen in the form of nanotheranostics, a portmanteau to encapsulate the multifunctionality of such nanoplatforms that can simultaneously provide targeted non-invasive disease imaging and drug therapy [11]. In spite of their remarkable potential for revolutionising medical treatment and contribution to the burgeoning of the personalised medicine treatment protocols, no such nanotheranostics have reached the clinic. Indeed, much of the available literature suggests that these have been developed as isolated efforts by numerous small academic research groups worldwide, and that there is a notable gap in the translational effort from “bench to bedside” [12]. As the Gartner hype cycle [13] would suggest, reaching the slope of enlightenment for nanotheranostics with their commercial realisation would require a redress of this gap, which would require improved regulatory frameworks for their development, more comprehensive understanding of their interaction with biological systems, demonstration of biocompatibility, and, most arguably, the development of predictive orthogonal models to illustrate in vitro and in vivo quality, safety and efficacy to reduce animal testing and pave the way for clinical development [14].

If clinicians have at their disposal in vitro models that can be used for high-throughput screening, more hits will inevitably be ascertained in a shorter timeframe. Furthermore, by utilising human-based tissues in their construction, robust biorelevant models are realised that recapitulate pivotal aspects of the BBB, thus dispensing with the requisite of cumbersome animal testing. The net outcome is accelerated development cycles, in which more attention can then be placed on the modularity and rational design of the nanotheranostic particles themselves for optimisation of brain-targeting delivery efficiency, which will be explored in a subsequent section. Due to the versatility of the models once appropriately characterised and validated, the researcher can focus primarily on optimising the delivery across the BBB, rather than exclusively considering the design of a particular nanotheranostic platform for one disease only.

The nanotheranostic platforms (NTPs) can generally be stratified on the basis of their constituent composition into organic, inorganic, metallic or carbon, but also sometimes more usefully grouped by their properties, and as such comprise the metallic, magnetic and semi-conducting NTPs. [15] This latter classification strategy is more useful in terms of providing an orthogonal comparison of such multifunctional carriers consisting of a core matrix, a diagnostic agent, a therapeutic agent and a tuneable surface of targeting moieties and polymeric coating for colloidal stability and conjugatable functional groups [16]. As these structures are typically in the range of 1 to 100 nm, they exhibit unique properties such as high surface area to volume ratio, enhanced permeability and retention, electrical and optical properties that do not apply at the macromolecular level [17]. This makes them viable for crossing the BBB as illustrated in Figure 1.

Brain delivery approaches have also focused on nasal administration and intracerebroventricular administration (ICVA), but these methods have thus far not garnered much traction due to deleterious effects and the fundamental issue that these drug delivery systems do not reach the primary requirement of releasing the drug in a steady state in a dose considered to be nominally therapeutic [18]. Indeed, ICVA in particular has been met with scrutiny due to the invasive nature of its delivery, and, as such, nanocarriers would seem an attractive and definitive alternative [19].

A considerable evidence base for these nanostructures has arisen, and numerous seminary review papers have aimed to consolidate such studies of nanotheranostics for brain delivery [7,8,11,20,21]. However, in many cases, the focus is on treatment efforts for specific disorders in isolation such as brain cancer, or indeed serve to appraise the critical quality attributes and behaviours of one nanomaterial in crossing the BBB. As such, the aim of this review is: (1) to summarise the BBB targeting strategies employed currently; (2) highlight new trends in rational nanotheranostic design for transport across the BBB; and finally (3) how various in vitro modelling techniques can lend themselves to abbreviated testing suites using less animals, particularly in relation to those that employ cells of human origin, as these constitute the gold standard in relation to predictive power and bio-relevancy.

Despite the promise of nanotherapeutics as a drug delivery strategy for CNS disorders, and some 250 papers published on PubMed in 2020 alone in relation to their use for crossing the BBB, as of this review there are only three nanomedicines licensed for use in brain cancer (Marqibo, Onivyde and Feraheme), with notable omissions of any licensed nanomedical treatment for Alzheimer’s disease (AD) and Parkinson’s disease (PD) [22]. This is bitterly disappointing in light of the intensive research efforts that have been focused on the application of nanomedicine to these fields in particular in recent years.

This review thus aims to demonstrate the diversity of targeting strategies for crossing the BBB with specific reference to nanotheranostics, identify the reasons for the lack of the clinical translation of research data generated thus far and identify trends and future directions for in vitro BBB permeability testing, which will reduce the need in future for reliance on animal studies. While numerous exemplary efforts on nanoparticle engineering have been reviewed, many have not extensively reviewed the possibility of in vitro modelling approaches for testing the merit of nanotheranostic candidates designed by formulation scientists. This review thus serves to inform formulation scientists and those working in the nanotechnology research area in relation to alternatives to in vivo administration for testing the synthesised nanoparticles in relation to delivery efficiency and indeed for establishing their biocompatibility and targeted delivery specifically to the brain across the BBB. As such, an appraisal of the interaction of nanoparticles with biological systems in relation to the formation of the protein corona, evasion of the mononuclear phagocytic system (MPS) and modulation of their physiochemical properties for enhanced colloidal stability, reduced clearance and increased accumulation by exploiting the EPR effect are beyond the scope of this review as they have been extensively reviewed elsewhere. [23,24,25] The specific interest of this paper is overcoming the BBB and how nanomaterials can be rationally designed and tested using modelling of the BBB to facilitate clinical translation of promising nanoplatforms, with abbreviated in vivo testing requirements.

## 2. Physiological Aspects of the Brain and Blood–Brain Barrier

The encephalon or brain is a complex organ that is responsible for regulating and integrating a complex array of executive functions in mammals including wakefulness, memory, sleep, olfactory signal integration, motor function and perception. It is, however, a considerably fragile organ, and therefore found enclosed in the cranium of the skull. Despite this physical enclosure to resist mechanical insult, it must further be physically protected from exposure to toxins and microorganisms present in the systemic circulation and the integrity of its physiological environment must be maintained. The BBB also acts as a strict barrier to the passage of xenobiotics, and, as a result, it is an aspect to circumvent to achieve adequate pharmaceutical delivery. In the brain, there are three such principal barriers: the blood–brain barrier (BBB), blood–leptomeningeal barrier (BLMB) and the blood–cerebrospinal fluid barrier (BCSFB), with the former constituting the key homeostatic regulator mediating transport between the peripheral circulation and the CNS [26].

The BBB is a cellular barrier constituted primarily by a concentric series of microvessel non-fenestrated continuous endothelial cells with tight junction adjoints, which was first described by Paul Ehrlich in 1885 [27]. This barrier thus exquisitely regulates the passage of xenobiotics, microorganisms and endogenous entities such as macrophages and endopeptides. It is increasingly acknowledged that the BBB is in fact far more complicated in composition, with contributions from several proteins (i.e., claudin-5 [28] and cell types (e.g., perivascular astrocytes), as the former provide high transendothelial potential resistances in the order of ~1500 Ohm/cm^2^ [29], thus significantly hampering paracellular mechanism of drug delivery, and the latter contribute to capillary phenotypic regulation.

This constitutes a significant hindrance to the development of therapeutic or diagnostic agents for brain-targeted drug delivery, as many agents may appear bioactive but cannot be permitted across this barrier. There are also a number of less tightly modulated regions such as the chemoreceptor trigger zones (CTZ), which regulate blood composition [30], primarily localised in the subfornical organ and organum vasculosum, which are in effect a dynamic permeable blood–brain barrier. However, these are less exploitable for therapeutic delivery, as they would generally impose dose limiting nausea and vomiting constraints, which are already a hallmark issue in the administration of intravenous chemotherapy.

This is further confounded by the presence of a robust biochemical barrier of efflux transporter proteins, which compromises the utility of the transcellular route for drug delivery. The ATP-binding cassette transporter efflux pumps (ABCs), multi-drug resistance proteins (ABCG2) and P-gp in particular are restrictive in facilitating antineoplastic and anti-human immunodeficiency virus pharmaceuticals delivery in conventional formulations [31]. The influx transporters such as the organic anionic transporters (OATs) are more exploitable as they can regulate transport in both directions, and efforts are directly at preferentially facilitating influx [32]. The blood–cerebrospinal fluid barrier (BCSFB) by contrast is not as popular a candidate in the drug delivery strategy for CNS disorders, owing to the arguments presented that this rather constitutes a principle aspect of ICVA as aforementioned. However, this highly branched structure of choroid plexus epithelial cells, which is responsible for homeostatic CSF secretion and regulation, can contribute to the delivery efficiency, as it too has a polarised expression of numerous ion channels, transporters and receptors. Thus, while acting as an essential protector and regulator for the brain itself, the BBB, BLMB and to a lesser extent the BCSFB pose a major issue to formulation scientists and academic researchers for drug delivery.

### 2.1. Modulating the BBB

A number of physical and chemical methods exist to modulate the integrity of the BBB in a temporary and reversible manner, using immune cells [33], techniques such as focused ultrasound [34] or more selectively using endogenous ligands as “Trojan Horses” [35], but these have not reached fruition clinically as such, due to issues with reproducibility and limitations with mapping biodistribution after administration. The neurosurgical methods or direct administration of such agents to the brain have been largely precluded except in experimental circumstances for the foregoing reasons of invasiveness, pain and risk of irreversible brain damage due to the unpredictability in the disruption to the BBB that ensues. [36,37] As such, the fundamental challenge remains in ensuring efficacious delivery across the BBB by the associated permeation mechanisms, and nanotheranostic delivery systems are an ideal candidate for such purposes when it can be demonstrated that they traverse the BBB and release their diagnostic and therapeutic payloads in a controlled and site specific manner [38]. These include passive diffusion, carrier-mediated transport and adsorptive or receptor mediated endocytosis/transcytosis.

### 2.2. Rationale for Nanotheranostics over Standard Therapies

The rationale for nanotheranostics in the treatment of neurological disease by crossing the BBB upon systemic administration is underpinned by the limitations of conventional empirical therapy [39]. Despite numerous advances in this field, particularly in relation to biotechnology and the revolution of monoclonal antibodies and immunotherapies, such therapeutics have hardly increased the delivery efficiency, nor significantly ameliorated the competence of current clinical diagnostics and chemotherapeutic regimens in particular [40,41]. One of the primary attributes of maladies of the brain is that prompt primary diagnosis treatment that is tailored to the patient is at a premium. Indeed, it would seem that nanotheranostics could potentially satisfy all of these pre-requisites and more, by limiting off site action, dose limiting toxicities and potentially overcoming drug resistance, which have significantly hampered clinical efforts thus far [42].

Acute traumatic insult to the head is commonly encountered in the clinic from multiple sources, most notably as a consequence of engaging in physical sport. This trauma to the head causes a multi-faceted pathophysiology to the BBB, including ischaemia, hypoxia, pro-inflammatory factor release and increased tight junction leakiness, which is potentially life threatening and for which there is not a satisfactory treatment to date [43]. A number of efforts including that by Campbell and colleagues [44] seek to identify such protocols for managing acute ischaemic head injuries, and nanotheranostics have potential in this regard [45]. To accelerate the progression of such to the clinic, however, there is an evident need for high throughput robust in vitro models to test such novel strategies, and the models outlined in this review hold great promise in this regard.

Despite this however, personalised medicine marks a paradigm shift in clinical models of care, and any such contributions to such should be heralded as a move away from the dogmatic stagnation of the neurotherapy field, in which superficially enhanced efficacy in vitro has been possibly given precedence over consideration of the in vivo translation and bio-compatibility of such delivery systems, which is the primary aim of any such exercise [46]. This underscores the potential of nanotheranostics in that the neurology and oncology fields are at a critical juncture as highlighted in a recent paper by Aldape and colleagues [47], in which they posset that current clinical diagnostics are arguably not sensitive enough for diagnosing such conditions at the pivotal sub-clinical stage. There is also an increasing recognition that the genomic, metabolomic and phenotypic heterogeneity of people is such that “one-fits all” empirical therapies are not optimal, and the incidence of such diseases are on the rise at a global population level [47].

This means that if clinicians have an armament of multimodal nanoformulations that can: (1) be specifically used to diagnose a patient, (2) stratify such patients according to likelihood of response by biomarker and metabolomic screening, (3) specifically and efficiently target the affected CNS tissues reproducibly using the full repertoire of pharmacological agents including biotechnological products, (4) monitor the biodistribution, tissues or response in real time and (5) follow-up and recalibrate the therapy based on response, this would mark a golden age of therapeutics. Optimism must be tempered by the fact, however, that as yet such “magic bullets” are confined to research settings, some promising clinical candidates and a handful of commercially approved agents that have one modality only, but concerted efforts towards such nanotheranostics will be reviewed forthwith in the specific context of systemic delivery targeting the brain via permeation across the BBB.

### 2.3. Rationale for Modelling the BBB

While small molecules <400 Da such as glucose can readily perfuse this dynamic barrier, in most cases a myriad of factors compromise the delivery of pharmacological agents [48], including lipophilicity, enzymatic degradation or metabolic conversion, association with non-transporting ligands or association with off-target tissues, and inefficient traversal to the interstitial spaces of the brain once in the parenchyma. The predictability of the in vivo efficacy of nanotheranostic carriers in crossing the BBB thus requires the development and implementation of robust models, which can reflect the dynamic nature of the BBB. This is further evidenced by the high attrition rate for candidate drugs (~80%) coupled with the fact that those that are therapeutically approved only cross the BBB in ~5% of cases [49].

Such in vitro and ex vivo models are indispensable at the pre-clinical stage, as the more extensive the information that can be garnered the less mandate there is for animal testing, which is costly and ethically contentious. This is in accordance with realising the “Replacement” component of the 3 R principles sanctioned by several national authorities outlined in the EU under Article 4 of EU Directive 2010/63/EU “on the protection of animals used for scientific purposes” [50].

The conventional studies of transfer across the BBB utilise membrane models, in vitro static and dynamic models or indeed animal studies, the latter of which has numerous established limitations in predicting in vivo behaviour due to inter-species variation [51]. It is no surprise that while as an academic exercise a lab mouse or their cells is useful for studying a novel nanoformulation for its BBB permeation properties, in effect, the translation of such to a human subject is virtually incomparable. As such, optimisation of in vitro models and permeability assays would constitute a move towards more comprehensive and predictive data, which would facilitate nanotheranostic candidate selection and optimisation at earlier stages for accelerated discovery and development.

### 2.4. Overview of Current Modelling Approaches

In vitro BBB modelling has been a subject of intensive research since the 1980′s, and in many cases has been based on the use of transwell assays, which although utilitarian have notable disadvantages [52]. They have been bolstered by the use of human induced pluripotent stem cells (iPSCs), which are more representative of in vivo conditions, but limitations associated with irregularities due to co-differentiation and the relative shortage of viable stem cell sources means that they have not been widely adopted to date [53]. As static monolayer models, they do not recognise the significant influence of shear stress on the endothelial function due to blood flow, and for this reason dynamic models have become more ubiquitous. Advances in microfluidic technology and integrated sensors has generated an organ-on-chip in vitro model of the BBB termed a μBBB, which can incorporate shear stress and can withstand high-throughput screening (HTS) [54].

Perhaps the most promising modelling approaches incorporate co-culturing of astrocytes, pericytes and primary brain endothelial cells to form CNS organoids as illustrated by Cho and colleagues, which will effectively be biomimetic of the neurovascular unit (NVU), as all of this constitutively contributes to BBB integrity [55]. The ease of culturing, up-scale for HTS and the fact that unlike conventional transwell systems all cells are in direct contact with one another means that these are a promising technology. They can also be rotated at regulated speeds to simulate the shear stress that microfluidic technologies provide, without additional expertise or specialised equipment considerations, and can be used to simultaneously investigate hundreds of compounds using automated robotic assisted confocal fluorescence microscopy and imaging mass spectroscopy [56]. The latter techniques in particular are an exciting prospect, as confocal fluorescence microscopy can be used for mapping biodistribution of nanotheranostics, which incorporate fluorescent dyes or quantum dots, while imaging mass spectroscopy facilitates detailed 3D imaging of the nanoparticles in situ for real-time clinical diagnostics and principal component analysis [57,58].

As evidenced by the foregoing, the selection of the BBB model is dependent on the intended purpose for conducting the study and the stage of development. The combination of such models with in silico screening technology would arguably lead to the elucidation of a greater number of viable leads and prediction of permeability and bio-fates at early stages of nanotheranostic development to save time and costs [59]. In terms of in silico screening of BBB permeation and ADMET parameters, the data selection process is imperative and must be extensively validated to minimise the false positive rate. However, recent advances in artificial intelligence and machine learning means that by producing initial robust classification models consisting of reliable nanocarriers, theoretically, the biased data sets can be corrected [60].

As a general rule, using different immortalised cell lines is usually warranted to serve as orthogonal models to qualify the in vitro to in vivo extrapolation of findings [61]. Signalling pathways and associated kinetics of transport are best suited to study by way of monolayers, as these are specific and simple [62]. For establishing structure activity relationships, and, more crucially, for evaluating toxicological profiles, more sensitive models such as iPSC models are warranted due to enhanced sensitivity and the critical nature of the information garnered in guiding subsequent optimisation of leads and generation of safety data as supporting information for clinical testing application submissions [63]. Organoids in particular would seem the optimal candidate for analysing nanotheranostics in tandem with organ-in-chip microfluidics, as they best recapitulate physiological conditions and integrity of the BBB and can be used to precisely determine cellular uptake and biodistribution in related high-throughput assays in a cost-effective manner. The relative strengths and weaknesses of such models are summarised in the graphic in Figure 2, which further exemplifies the increase in choice with respect to time as more technologies come on stream [54].

## 3. NTP Delivery Approaches for Treating CNS Disorders

Of the aforementioned mechanisms of transport across the BBB, adsorptive mediated and receptor mediated endocytosis constitute the most pervasive explored by researchers for NTP mediated delivery of imaging contrast agents and therapeutic moieties. These biological mediated mechanisms and to a lesser extent cell mediated delivery will be the focus of this review in relation to testing the efficiency of NTP delivery across the BBB.

A summary of the main advantages and limitations of the various strategies employed by nanoformulation scientists for brain-targeted NTP delivery are summarised in Table 1. When the contemporary literature is investigated, the most promising and readily tested NTP platforms are those that make use of surface functionalisation with either known ligands of the receptors highly expressed on BBB endothelial cell surface, or indeed by using inherent cellular components such as macrophages [35] and fatty acids to enhance penetrance. The key factor is to determine not only whether the NTP can be delivered across the BBB model in-vitro, but also to have a measurable index of the concentration or number of particles that reach the brain (as well as accumulation in non-target tissues), as this is the true predictor of therapeutic response [7,8] and biocompatibility.

As alluded to, these constitute the non-invasive branch of brain delivery technologies, and the invasive technologies are thus beyond the scope of this review [64]. Comprehensive reviews of same can be consulted if necessitated, as these too need adequate models of the BBB to investigate the viability of such grafting and direct injection on the integrity of brain tissues and evaluation of safety and toxicity, particularly with repeated administration [68,69].

### 3.1. Adsorptive Endocytosis-Mediated NTP Delivery

Adsorptive-mediated delivery to the brain involves the functionalisation of the NTP surface with cationic components to selectively target the net anionic surface charge of luminal surface of endothelial cells of brain capillary [70]. This charge is a consequence of clathrin vesicles that function to regulate ionic trafficking of molecules, and to specifically repel anionic species. The most promising examples constitute those employing cationic bovine serum albumin (CBSA) and trans-activating transcription (TAT) peptides on the surface [71,72]. The size and surface charge is tailored to preferentially facilitate association, with subsequent engulfment and exocytosis towards the abluminal surface of these cells.

They can also be used to condense and bind nucleic acid, and thus have been demonstrated to successfully deliver DNA plasmid across the BBB to brain tissues. A number of candidate neuropharmaceuticals, including gene therapies, neuroprotective agents and chemotherapeutics, with enhanced permeability, were confirmed by images to have increased accumulation, and in some cases sustained release profiles [73,74]. The most viable materials for such platforms are pegylated chitosan, lipid and polymeric nanoparticles such as polylactic acid (PLA), poly-Ɛ-caprolactone (PCL), cholesterol, poly(butyl cyanoacrylate) (PBCA) gelatin siloxane and mesoporous silica magnetic nanoparticles incorporating iron oxide (SiO_2_-Fe_3_O_4_) [75,76]. Some isolated instances of glutathione and sinapic acid-based as well as MMP-2200 derivative functionalisation have also been investigated to remarkable results [77].

However, the primary issue with this class of NTPs is that despite their potential they are notably more toxic than non-ionic or anionic counterparts, which must be appraised before recommending their scale-up and clinical testing. A paper published by Lv and colleagues elucidated such structure–toxicity relationships, and determined that for such non-viral vectors, low molecular weight polymers such as PLA and PLGA are preferable, and the biodegradability of the linker is pivotal [78]. In such cases, a carbamate linker is preferred where viable, and for cationic lipids importing a heterocylic ring as the head group in preference to a quaternary or tertiary amine is preferred. It is also purported and clarified with reference to more contemporary literature that engineering of self-assembling amphiphilic carriers or water soluble lipopolymers including those based on poly(ethylenimine) (PEI) and poly(l-lysine) (PLL) and non-ionic actively targeted “niosomes” are the best strategies in relation to gene delivery in particular, which has notable implications in several CNS disorders including Huntington’s disease, AD, PD and glioblastoma multiforme (GBM) [64,79,80].

### 3.2. Receptor-Mediated Transcytosis

In keeping with the marked trend away from nanomedicines being designed and tested based primarily on the enhanced permeability and retention effect (EPR), which has notable limitations particularly with regards to the heterogeneity of response and lack of reproducibility in vivo, perhaps active targeting utilising functionalised receptor ligands for active targeting is the most promising strategy for the novel nanotherapy driven drug delivery systems. This is unsurprising given the exquisite regulatory function of the BBB and associated biochemical barriers to entry of exogenous compounds. As a direct consequence, by employing ligands that preferentially bind the iron transferrin, folate, insulin, and LDL cholesterol receptors, among others that have been studied, and predictable pathways of internalisation to the brain, it can be appreciated that these are the most probable candidates, particularly when exploited synergistically [64].

#### 3.2.1. Transferrin (TfR) Receptor-Mediated Transcytosis

The most widely studied of the foregoing is arguably the iron transferrin (TfR) receptor, as they are very highly expressed in the brain endothelium in comparison to the periphery, although the bone marrow, splenic and hepatocellular accumulation is always a concern [81]. The lactoferrin receptor is also a notable member of this family and has been targeted to varying success in some instances, such as that achieved by Kumari and colleagues for temozolomide delivery, which was demonstrated both in vitro and in vivo to improve its pharmacokinetics and intratumoral accumulation by a pH-dependent responsive mechanism [82].

A number of notable achievements have been made by careful optimisation of the physiochemical composition of such nanocarriers, as it became increasingly evident that naked nanoparticles >200 nm would not garner a suitable therapeutic concentration due to efflux and the requirement for recycling before selective accumulation in brain tissues. A number of immunoliposomes have been developed using antibodies such as OX26, which recognise alternative epitopes on the transferrin receptor, as illustrated by Kang and colleagues for dopamine delivery in a rat model of PD, achieving an 8-fold increased uptake compared to naked dopamine and 3-fold compared to pegylated liposome alone [83]. Such immunoliposomes achieve this enhanced delivery by occupying these alternative epitopic sites as the receptors are usually saturated in a physiological condition with endogenous protein.

This has been achieved to considerable success with gold nanoparticles (AuNPs), folate and transferrin dual conjugated doxorubicin loaded liposomes for glioma treatment as demonstrated by Gao and colleagues, which can be further modulated to incorporate imaging agents, paclitaxel, cisplatin and other notable therapeutic payloads such as amyloid β-inhibitors and siRNA [84,85,86]. The most notable requirement seems to be that antibody targeted carriers require monovalent antibodies with carefully tailored affinities such that the antibody does not bind too strongly and result in the receptor complex being phagocytosed [87]. The prototypical example in this class would be JR-141 (Pabinafusp Alfa), which was recently approved in Japan for the treatment of Hunter’s syndrome (mucopolysaccharidosis II, a rare heritable carbohydrate storage disease) [88]. JCR pharmaceuticals have patented a proprietary BBB permeating technology “J-Brain Cargo”, which utilises a fusion protein comprising an anti-TfR antibody and iduronate-2-sulfatase as an intravenous enzyme replacement therapy.

Despite its orphan designation in Japan, which was approved in March 2021, this constitutes a major breakthrough for such platforms, as this proprietary modular platform can be potentially used for brain-targeted delivery for other diseases, such as mucopolysaccharidosis I, which is being evaluated using JR-171, a fusion protein of J-Brain Cargo and α-L-iduronidase (IDUA) [89]. Although its inclusion is on the basis that antibodies are essentially nanomedicines in their own right, it exemplifies the promise of such receptor-mediated delivery systems, with the primary consideration for testing ensuring the model accounts for the inter-species TfR expression disparities (2.5-fold higher in mice brain microvessels) [90]. This again demonstrates that predictive BBB models need to be sophisticated enough to account for such nuances, but may be preferential to resorting to using human TfR knock in mice, which must also account for receptor “sinks” of peripheral compartments potentially influencing the overall therapeutic concentration at target tissues.

#### 3.2.2. Low-Density Lipoprotein (LDL) Receptor-Mediated Transcytosis

The low-density liprotein (LDL) gene family have crucial contributions to regulation of metabolism and nutrient transport in mammals, and this holds true for the CNS, particularly in relation to apolipoprotein E (apoE) [91]. ApoE is synthesised by microglia and astroglia, and it has been suggested increasingly that it has a role as a susceptibility gene for AD and contributes to the neurobiology of disease following such insults in immunomodulatory and neurotrophic as well as antioxidant contexts [92]. This gives an inherent degree of versatility to the construction of nanocarriers for these receptors, as a number of endogenous compounds can be used as biomimetic scaffolds for high-throughput screening. Solid lipid nanoparticles are the most widely employed carrier class in this regard, although there has been notable disparity in terms of their success in permeating the BBB, which potentially is a consequence of certain nanocarrier properties imparting an adsorptive-mediated transcytosis mechanism preference over directly using the lipoprotein receptor related protein (LRP) ligands [93,94].

The angiopep 2-based ligand in particular has a notable dual targeting functionality which can be modelled in vitro for predictive response, as this ligand is expressed on glioma and amyloid β cell surface as well as on the BBB. As a result, it enhances accumulation in the brain by receptor-mediated transcytosis, and successively facilitates localisation to such disordered tissues for mediating a clinical response, which has been demonstrated by Kafa and colleagues who employed targeted nanotubes in glioma in vitro and in vivo models [95]. The BBB model of porcine brain endothelial cells (PBEC) co-cultured with rat astrocytes demonstrated diameter dependent accumulation at 24 h of approximately 2% of the injected dose/g brain. The natural HDL carriers are perhaps even more desirable due to their enhanced stability, biocompatibility and long circulation with intrinsic biological function properties, as intravenous administration of apolioprotein A1 nanoparticles alone have reduced amyloid β levels in symptomatic APP/PS1 mice models for AD.

Both direct conjugation of apolipoproteins and indirect methods which employ non-ionic surfactants such as the polysorbates to promote subsequent apolipoprotein adsorption in vivo have been explored. The literature seems to find agreement in the fact that administration route has another critical determinant influence on the efficiency of such formulations, with pulmonary administration intriguingly leading to higher effective brain concentrations of the nanoparticles when compared with intraperitoneal and intravenous administration, though again one must consider the extrapolation of such data from mouse to human models of the BBB [96,97].

One notable limitation is the availability of primary LDL ligand materials, and as such mimetics employ materials such as acrylic polymers, i.e., PBCA, phosphatidylcholine, triglycerides and PLGA surface functionalised with Tweens and Spans, as well as more contemporaneously with angio-pep 2-based ligand, they have been employed with both in vitro and in vivo successes. Costagliola di Polidoro and colleagues [98] designed hyaluronic acid nanoparticles encapsulating an imaging agent (i.e., Gadolinium--diethylenetriamine penta-acetic acid) and irinotecan, which when surface functionalised with angio-pep 2 led to improved glioma imaging through enhanced T_1_ relaxometric properties and cytotoxic efficacy at 24 h rather than 48 h, thus reducing irinotecan time response. These have also explored tentative use of the oral route, which would be considered the gold standard of administration routes due to acceptability and tolerability for the patient. Dalargin, an anti-nociceptive peptide mimicking endogenous opioid peptides was successfully found to localise in the brain endothelium following oral administration in a PBCA nanoparticle formulation surface coated with Tween-80 [67].

#### 3.2.3. Other Notable Receptor-Mediated Approaches

Proteomic studies have generated invaluable information in relation to the endogenous regulation of the BBB and have recognised several other receptors that can potentially be commandeered by nanomaterials for passage into the brain [97]. For example, studies of models of epilepsy have revealed that glutamate in particular can modulate in vivo BBB permeability, and as it is recognised by several receptors and is implicated in several disorders, i.e., anxiety, epilepsy, pain and addiction, this means that it holds noteworthy promise [99]. The glucose receptor (GLUT-1) is upregulated in brain tumours due to the hypoxic environment and may be an associated marker of radio-resistance and poor prognosis [100]. Additionally, a rapid glycemic increase is observed following fasting which has been demonstrated by Wu and colleagues to impart rapid delivery character to a number of nanomaterials including micelles, both in vitro and in vivo in models of head and neck squamous cell carcinoma (HSNCC) [101].

While insulin cannot itself be readily employed for mediated passage of the BBB due to instability of the endogenous ligand and hypoglycemic potentiation, anti-insulin receptor antibodies have successfully been conjugated to nanocarriers for active targeting of brain tissues [102]. Ulbrich and colleagues provide an eminent example of such a strategy employing 29B4 anti-insulin conjugated loperamide loaded human serum albumin nanoparticles versus immunoglobulin G conjugated nanoparticles in an antinociceptive tail flick test in ICR (CD-1) mice [103]. The fact that the latter had only marginal effectiveness demonstrates the potential for anti-insulin antibodies in considerably increasing the delivery efficiency. EGFR, folate and, more recently, interleukin receptors have been implicated in cancer due to their high expression on tumour cell surfaces, and have been studies as a targeting mechanism for several years [104,105]. Peptides, magnetic nanoparticles and quantum dots have all been successfully used for enhanced chemotherapeutic and imaging applications by selective recognition as the folate receptor in particular is highly expressed on the BBB but not on healthy brain cells, and, as such, a dual targeting efficiency is achieved both in terms of facilitating passage across the BBB and further in localising to tumour tissues.

Cai and colleagues successfully designed and tested a nanotheranostic platform consisting of an aggregation induced emission fluorogen for glioblastoma multiforme tumour margin imaging and a high NIR absorbing semi conducting polymer for successive photothermal therapy encapsulated in cRGD and folate surface functionalised nanoparticles. [106]. These nanoparticles had good biocompatibility and safety demonstrated by almost complete clearance at 10 days, and, furthermore, the optical properties facilitated vivid tumour size analysis up to a week following tumour implantation and offer selective GBM cell killing efficiency.

Another robust example is the EGFR variant III targeted by Peng and colleagues using aptamer U2-gold nanoparticle complexes (U2-AUNPs), constituting a novel and promising strategy for GBM treatment [107]. In both the in vitro U87 cell line and in tumour bearing mice, significant antitumour efficacy was observed (effectively halving the percentage of proliferating cells when treated with U2-AUNPs, versus a negligible response for AUNPs alone), and increasing survival times of treated mice (mean 30 days versus 24 days for those treated with the NaCl control). While unlike Cai and colleagues this study does not significantly address safety concerns of using such AUNPs, what is evident is that EGFR targeting is a viable strategy for treatment of gliomas by selectively inhibiting the associated proliferation and DNA repair pathways.

### 3.3. Other Active Targeting Strategies

The foregoing notable advances in this field serve as a concise demonstration of the versatility and utility of rationally designed nanocarriers, which include various modular structures, surface chemistries and formulation with the emergence of nano-emulsions, in situ nanogels and self-assembling nanosuspensions [108]. In general such strategies make fortuitous use of the fact that the neuropathophysiology of glioma, AD and PD among other neurological disorders including ischaemia and acute neurological trauma involves an innate disruption of the integrity of the BBB due to neuroinflammation and dysregulation resulting in increased permeability [109]. Focused ultrasound has garnered attention for synergistic therapies involving intravenous administration of ultrasound sensitive microbubble nanoformulations followed by ultrasound-guided temporary opening of the BBB [110]. This facilitates temporary reversible increased site-specific permeability changes for subsequent administration of nanoparticles, imaging agents and cells, which has shown particularly promising results for magnetically guided superparamagnetic iron oxide nanoparticles (SPIONs); the safety of such approaches remains dubious [111]. These consolidated non-invasive strategies are perhaps best conceptualised by visual representation as given by Figure 3 [112].

Where the BBB is in its intact physiological state, however, more exquisite strategies are required, such as active peptide sequence targeting, i.e., using iRGD for BBB and tumour penetration enhancement [113]. A number of shuttle peptides have been developed as a consequence of improvements in phage display technology, and cell-based transportation technologies such as those highlighted by Li and colleagues and Batrakova and colleagues, respectively, are propitious, despite admitted limitations associated with heterogenous expression and limited loading capacities [114,115]. These approaches consequentially must also be accounted for when designing in vitro models of the BBB, as safety is the paramount concern, particularly for inorganic nanoparticles employing heavy metals or non-biodegradable moieties, despite their useful optical and magnetic properties [116]. The vast array of nanoparticulate systems in terms of design, materials and modulation in terms of therapeutic, diagnostic imaging agents and surface probes with divergent biodistributions and principal activities require a parallel robust toolset of viable predictive models to test their effectiveness.

## 4. Towards Consolidated NTP Testing Using Validated BBB Models

If safety and biocompatibility can be unequivocally proven, then more liberal regulatory frameworks with abbreviated testing protocols would pave the way for accelerated development and approval. It would not seem useful to design an intact BBB for testing NTPs destined to be used in pathological states, but, by the same token, designing a dysfunctional BBB may artificially lead to results constituting effective permeability of such nanocarriers when in fact this would not be clinically reproducible. Thus dynamic models are the gold standard, which are feasible due to improvements in microfluidics, cell engineering in tandem with in silico screening technological capability advancements which have been witnessed in the last decade.

As alluded to in Section 2.3 and Section 2.4, the ultimate goal in research and development is to find universally acceptable and applicable in vitro BBB models that essentially recreate the neurovascular unit, as shown in Figure 4, in order to expedite research and development and reduce the associated financial and logistical implications of using animal testing as the primary source of supporting clinical information. Furthermore, they hold more constitutive properties when they can mimic physiological condition such as receptor expression, cellular regulation and stresses such as shear stress due to blood flow, which can then be used to rapidly evaluate a wide range of nanomaterials and nanocarrier platforms for their permeability efficiency. Such models are preferential to conducting in vivo studies on animals, and the trend of their development and increasing use by researchers is chronologically reviewed in a seminal paper published by Ribeiro and colleagues [117].

These include monolayer isolated brain capillary models, in vitro cell-based models using human and animal-derived cells and cell-free models including microfluidic “brain on chip” models, which all have merit and associated challenges and limitations in relation to their application to studying nanomaterials. These are able in an orthogonal manner to account for such nuances and heterogeneity and hold promise for reducing in vivo testing studies to prove their merit, which would be a remarkable achievement in the context of regulatory and drug development models. As they cannot entirely reproduce the in vivo environment, knowing the limitations of a model or cell type in advance can be pivotal in governing their selection. The merits and challenges constituted by such models which will be discussed in detail in the next subsections in the context of their applications to NTP testing.

### 4.1. Validation Markers for the Reviewed Models

While it is generally considered to be practically impossible to generate a full set of BBB characteristics to ensure the models recapitulate all features of the barrier, a number of key parameters aid in ensuring the model is suitable for its intended study application. A seminary paper published by Helms and colleagues should be consulted for in-depth guidelines on protocols for the general use of these models [118]. While there are several established sources of heterogeneity in any in vitro cell-based model study, and reproducibility can be difficult, the lack of translatability of data is frequently due to incomplete characterisation of the models, nanomaterials and due to suboptimal handling and protocols for their use [119]. The following therefore constitutes an effective user guide for researchers in validating a model for the study of nanomaterials to ensure more robust data are generated, which will be more representative of the in vivo situation as presented in Table 2 [120,121,122,123,124,125,126,127,128,129,130,131,132,133,134,135,136,137,138,139,140].

A notable inclusion is the junctional tightness, as this arguably is the pivotal property that will influence the robustness of the model in mimicking the integrity of the BBB [141]. In the vast majority of studies, the tightness is measured using the transendothelial electrical resistance (TEER), which, while useful, has notable implications with its use, including differences in the techniques and apparatus used to measure it, and the size dependence of the compound of interest is largely ignored [142]. As such, permeability using hydrophilic tracers such as fluorescent probes or small molecules such as sucrose (~340 Da) may provide more functional estimates, particularly when both measures are employed, in addition to ensuring claudins are also included in the model to prevent BBB model leakage [143].

Expression and localisation of the efflux transporters and solute receptors outlined in Section 3 are also fundamentally important to the study, and particular care must be exercised when using non-human in vitro models. It has been established by quantitative-targeted absolute proteomic (QTAP) studies that the human BBB is closest to that of the cynomolgus monkey and marmoset primates in terms of receptor expression, and the poor efficiency of rodents is such that in most cases both receptor and efflux transporter expression is greater than two-fold higher in such model organisms [144]. For example, P-gp expression has been found to be expressed in the region of ~6.00 fmol/μg total protein in humans, but the expression is ~14 fmol/μg total protein and ~19 fmol/μg total protein for mouse and rat, respectively, which has been validated using positron emission tomography (PET) studies of the permeability of known P-gp substrates, which are far higher in human models due to lower function of the P-gp efflux mechanism.

Conversely, studies of claudins have elucidated that claudin-5 is the critical protein for tight juncture closure in humans and is two-fold higher than in other primates and rats, while mice have 1000 fold expression to that of humans [145]. This complicates the ability of certain models constituted by cells from rodent origin to be the most efficient for translational research efforts. However, as outlined by Ohtsuki and colleagues [146], exogenous expression of such proteins is possible with subsequent transfection to the rodent cell model without adverse effects, which constitutes a pragmatic workaround, and as will be outlined due to issues relating to cost and sourcing of human-based models, the in vitro animal models will continue to be imperative in the overall research and development process for nanotheranostics into the future.

### 4.2. Cell Culture Models

#### 4.2.1. Monolayer Cell Culture

The simplest models for the study of nanoparticle interaction with the BBB involve the culturing of primary endothelial cells on a transwell insert [53], which creates a two-compartment model [Figure 5] in which the insert mimics the luminal side (blood compartment) and the well mimics the abluminal side (parenchymal space). Although these primary cell lines are preferred due to their high TEER values (500–800 Ω cm^2^), high tight junction expression and classic BBB receptor and enzymatic expression (such as claudin-5, P-gp and occludin), the task of isolating these cells directly before cell culturing is an arduous task [147]. Indeed, for the mouse, such vasculature accounts for 0.1% *v*/*v* of the overall murine brain, which has a high propensity for contamination and additionally requires a large number of rodents to conduct one experiment.

Once a confluent monolayer general forms (after approximately 5 days for most studies), the model should be validated with fluorescence staining and appropriate TEER measurements and permeability assay to show adequate tightness. The use of a microporous membrane (~0.3 µm) support, usually of polycarbonate or polyethylene terephthalate construction, is obviated for nanoparticle permeation studies as it facilitates the passage of small molecules while maintaining the two-compartment model [148]. However, a possible exception is illustrated by De Jong and colleagues [137], in that the quantification of transendothelial delivery can be accurately determined by using a collagen gel covered with a confluent monolayer and is more accurate as association within the filter and membrane pores is eradicated.

The use of immortalised cell lines (bEnd3 cells) is beneficial as these are commercially available at a relatively low cost and circumvent the need for sophisticated isolation and cell treatment protocols, though limitations are implicated in their use in relation to reduced tightness of the monolayer formed [149]. Such models as evidenced by the studies are invaluable for high-throughput screening and studying transport kinetics and elucidating the permeability pathway of diverse nanocarriers for both passive and active targeting strategies. However, as only one cell type is employed, it does not satisfactorily address key aspects of the NVU, and, consequently, such studies are limited in their use for biocompatibility and translational efficacy studies.

As outlined, the use of murine cell lines is complicated by the need to sacrifice many rodents to obtain a sufficient amount of the endothelial cells. To ameliorate this problem, several non-rodent and primate models [150] as well as human cell lines can be used, and one of the most eminent examples for the study of nanoparticles is porcine cells and human CMEC/D3 cell lines. As evidenced by the models used in the studies used in this review, a diverse range of nanomaterials can be studied for permeability and biocompatibility. The culture conditions are relatively similar throughout, with more robust models being generated by removal of the serum and the inclusion of several agents and conditions that favourably increase the tightness of the model. These include cAMP modulators and puromycin with hydrocortisone, as conducted by Teow and colleagues [127], which additionally facilitates interfacing with analytical methodologies such as HPLC for quantitative modes, as well as mimicking shear stress using dynamic model apparatus such as hollow fibre cartridges.

The limitations of porcine models are also exemplified by these studies in that for orthogonal comparisons, porcine in vivo models are less well characterised and readily studied due to the handling of such larger organisms in research settings. To support clinical studies and the translational significance of such studies in porcine models, co-culturing and extensive biocompatibility studies [151] in such in vitro models should be considered at the earliest lead optimisation stages to support the clinical significance of such gathered data.

The strengths of porcine models lie in the fact that such studies are relatively simple and inexpensive to conduct due to the large quantities of endothelial cells that can be isolated, cultured and cryopreserved, with subsequent rapid thawing and culturing for producing confluent monolayers [152]. This can contribute significantly in early nanotheranostic research and development to limit the financial, environmental and ethical implications of cytotoxicity and biocompatibility studies in research settings, potentially expediting the subsequent stages of development by providing robust translational safety data.

It is possibly no coincidence that the most exemplary efforts in transwell models are constituted by those which employ human-derived cells, as evidently these will have the most predictive power in mimicking several aspects of the in vivo dynamic BBB environment. The human brain microvascular endothelial hCMEC/D3 cell line is the most intensively studied and optimised monolayer model to date [153], with several notable features, including extensive characterisation of receptor, enzyme and tight junction proteins, but suffers from having lower TEER and permeability than other comparative models. These are most useful for biocompatibility and biodistribution studies as they have been interfaced synergistically with sophisticated imaging such as those observed by De Jong and colleagues [137], which facilitate real time imaging of translocation, in addition to association within the model itself for rapid analysis of nanoparticle–BBB interactions.

The primary limitation of human models such as these are the obvious ethical implications of resourcing human tissue and the relative paucity thereof, although this has been partially offset by the increasing cell banking and commercial availability of same. In addition to this, resourceful researchers have developed efficient and well characterised monolayer cell cultures from other sources: stem cell lines to yield induced pluripotent stem cells (iPSC) [119], as well as non-cerebral origin-derived materials such as the human immortalised epithelial colorectal adenocarcinoma cell line (Caco-2) [127] and CSF-derived human capillary choroid plexus endothelial cells [130] have all been employed successfully for studying nanoparticle bio-interactions.

For all the foregoing monolayer models, the fundamental commonality that precludes their use for predictive permeation studies and biocompatibility studies of nanoparticle BBB integrity disruption is that they employ one cell type only. Despite the fact that while these monolayer models are utilitarian in the alluded to instances, they are a gross simplification of the NVU, and as they are 2D models, they do not fully depict the anatomic structure and complexity of the BBB in vivo.

Efforts have been made to better express these 2D models in mathematical terms by Zhang and colleagues [137], producing a transcellular model that can efficiently monitor the determinant influence of surface charge, medium ionic concentration and viscoelastic properties on nanoparticle permeability, but these do not fully capture the physiological relevancy and otherwise rely on assumptions of the equations employed. Various cell lines have been up-regulated to maintain endothelial cell relevancy following isolation by activation of canonical pathways, such as the WnT/β-catenin pathway, to promote phenotypic behaviour [154], but they do not adequately represent the endothelial cell–astrocyte crosstalk which can be better recreated by co-culturing.

#### 4.2.2. Co-Cultured Cell Cultures

As alluded to, the primary goal of an in vitro cell-based model is to recreate as closely as possible the in situ BBB environment and composition of the NVU in terms of tightness and expression of transporters, as well as the key cross-regulatory and vesicular trafficking functions of endothelial cells, which are primarily modulated by the astrocytes and pericytes [155]. If such models are to essentially replace the classical in vivo methodologies for quantitatively assessing CNS permeability of drug candidates that encompass in situ perfusion, CSF sampling and intracerebral microdialysis, as well as intravenous and intraperitoneal, as less invasive methods [156], then in the wider context of CNS drug discovery programs, the models must be more representative and maintainable over longer study periods. To this end, co-culturing and indeed tri-culturing afford opportunities in this regard in maintaining the versatility, usability and high-throughput strengths of a monolayer model while consolidating their relevance to the in vivo situation.

While porcine and bovine sources of endothelial cells are advantageous for small molecule permeability studies such as nanocarrier BBB passage screening, the rodent models are possibly more representative due to their closer homology to human protein expression [97], with the implied compromises as alluded to in Section 4.1. While selected studies have employed a co-culturing protocol, with improved functional tightness (correlated by TEER measurements and reduced P_app_ measurements), as Hatherell and colleagues elucidated [157], direct cell-to-cell contact in such models is a prerequisite for transitioning from 2D to 3D models, and astrocytes have a greater contribution to their tightness than pericytes. This was demonstrated by the physical constraints of transwell systems, in that adjacent co-cultured cells are generally unfeasible, and the endothelial cells are instead seeded on the insert surface while the astrocytes are cultured underneath and the pericytes at the bottom of the apparatus. The result is that the intercellular communication is mediated only by soluble factors secreted into the medium, which is virtually impossible to characterise and reproduce with any study-to-study homogeneity.

What Gromnicova and colleagues remarked, however, is that this geometrical problem can be overcome by employing a 3D collagen gel under the confluent endothelial monolayer [126]. This is further improved by the commercial availability of such co-culture systems, such as that employed by Hanada and colleagues [132]. The co-culture systems have a dual functionality of permitting rapid permeability studies and more comprehensive biocompatibility assessment of “nano-risk”, as it can also encompass the impact of nanomaterials on astrocytes which they encounter directly after passage across the BBB. As astrocytes provide metabolic nutrients to neurons and are also neuroprotective, a fundamental aspect of the compatibility of certain materials can be assessed by investigating the potential impact of these on the viability of the co-culture model, which, as Xu and colleagues found, can consolidate the contradicting data in more rudimental studies in the broader literature [135].

While the vast majority of studies favour models which utilise rat-derived cells, there is an increasing appreciation of the viability of employing synergistic mammal and rat-derived models [133]. One such model can be envisaged to include the use of human-derived cells or neurons in place of pericytes for modelling the blood–CSF barrier in addition to the BBB. This would facilitate more complete in vivo correlations and extrapolation to humans using a triple culture model, which offers the highest functional tightness and biorelevant properties [158]. The expertise required and laborious nature of such models is a detrimental facet of their use, but with increasing sourcing of patented models, these frozen ready to use kits may possess the answers to issues raised by isolated efforts published to date in terms of reproducibility and characterisation of these models for the study of nanomaterials.

Co-culture models are also superior for studying CNS conditions such as stroke and traumatic brain injury, as Neuhaus and colleagues [159] have highlighted the immense influence of astrocytes on the in vitro BBB model they developed. Such conditions require biorelevant models for investigating underlying pathophysiological mechanisms and testing the potential merit of therapeutic strategies for their acute management. As was also alluded to, however, as for all transwell-based cell culture models, the contribution of shear stress by blood flow to regulation of the endothelial cell layer cannot be understated, and so dynamic models of the BBB in 3D are desirable. Takeshita and colleagues [160] have developed this concept to realising a human cell-derived co-culture model that simulates the shear stress under flow and, furthermore, allows recovery of the analyte after transmigration. While these evidently have great suitability and bio-relevancy, the discriminant contribution of various satellite cells such as neurons and microglia are not readily emulated, and in many cases, the study is confined to a 2D nanoparticle bio interaction study. Thus, while pivotal in R&D for high-throughput screening, lead optimisation, permeability studies and toxicity screening for functioning as supportive data for clinical testing applications in lieu of in vivo animal studies, testing on a more complex model is mandatory as an attractive and satisfactory alternative from a research and regulatory standpoint.

#### 4.2.3. Spheroid Cell Culture

With the foregoing considerations of the limitations of 2D models and the interaction of nanoparticles with dyes and artificially enhanced permeability leading to biased results, 3D cell culture models have garnered attention in modern times. The marked shift towards the use of such models has also been a direct result of the increasing recognition of the numerous advantages of induced pluripotent stem cells (iPSCs), both in terms of close physiological relevancy, reproducibility, scalability and isogenic and individualised co-culturing protocols facilitating patient-specific integrated CNS models [161].

Perhaps the most exciting prospect of such 3D models is that they can spontaneously self-assemble to form scaffold free models of the BBB for permeability screening, i.e., spheroids, which accurately represent the brain physiologically and spatially [162,163]. More sophisticatedly, cells can form cerebral organoids in suitable scaffolding matrices, such as that developed by Nzou and colleagues [140]. For an excellent review of iPSC-derived BBB models including cerebral organoids for studying neurological disorders, consult the article recently published by Logan and colleagues [164] in comprehensive physiology. While there is some degree of contention with regards to defining the differences between a “spheroid” and ”organoid,” what can be agreed upon generally, and for the purposes of this review, is that the difference lies primarily in the cell types used and the culturing protocols, and will accordingly be discussed separately.

Generally speaking, they share a commonality in their self-assembly and the fact that they essentially are an organ mimetic with shared characteristics to the endogenous organ [141]. Where they differ, however, is in the employment of differentiated or stem cells, which give rise to spheroids and organoids respectively. Therefore, in this section, the nanomedicine studies for cell culture are confined to spheroids/assembloids, and organoids will be further discussed in the section on organ-on-chip microfluidics, in how they can be used in tandem with microfluidics for studying nanoparticle–BBB interactions as an alternative to in vitro cell cultures. As was shown by the findings of Sokolova and colleagues [139], Nzou and colleagues [139], and Kumarasamy and colleagues [140], the most robust models must as a prerequisite include the primary elements of the NVU (endothelial cells, astrocytes and pericytes), but further yet, the necessity of the addition of satellite cells such as the microglia to spheroid models, the so-termed “third element” of the NVU by Szepesi and colleagues [164], which accounts for 10–15% of total brain cells, is incontrovertible.

To date, many studies have used the triple culture model only, and while this is generally a pragmatic compromise given the cost and technical constraints associated with sourcing, characterising, and culturing such auxiliary cells for low-adherence spheroid self-assembly, their importance in models for studying nanomaterials cannot be understated [165]. Indeed, Kumarasamy and colleagues [166] have extensively investigated the various contributions of microglia in mediating nose to brain transport of nanomaterials, an alternative nanotheranostic delivery strategy under consideration, and indeed the possibility exists of pathogens such as SARS- CoV-2 hijacking this mechanism for CNS entry [167].^,^ Indeed, where such models are obviated but cost considerations would preclude their use for small research groups, rather than resorting to simplified models with little predictive power, the use of a human-based triple culture supplemented with rat microglia and neurons would be the most advisable approach.

The strengths of such a model are best represented in the fact that dyes themselves will not enter the model, but when encapsulated in a nanocarrier, advanced microscopy techniques such as confocal laser scanning microscopy can be used to map real-time permeation and biodistribution in a vascularised model of the BBB. [138,140] As this is spatially resolved in highly resolved 3D audio-visual recordings, and the fact that the model is more relevant than transwell culture models in terms of key modulator and receptor expression and activity, means that these are versatile and powerful models for drug screening [162]. In addition to this, pathophysiological states can be exquisitely modelled, and as Sokolova and colleagues exemplified [138], this is a noteworthy trait as nanomaterial-based management of hypoxia in acute phase ischaemic stroke and traumatic brain injury can be investigated.

The main limitation is that although the model can be extensively characterised and validated, and an array of molecules can be studied for their cell penetrating, receptor-mediated or other associated transport mechanisms of delivery across the BBB, the influence of shear stress is difficult to model unless the model can be extensively vascularised [168], which eludes all but the most experienced researchers. As such, advances in microfluidics can be used to solve this problem and generate an organ-on-chip, which can simulate the mechanics and physiology of the human brain in a micro physiological artificial organ system [54].

## 5. Recent Trends and Future Directions for NTP Models

### 5.1. Microfluidic Organ-on-Chip Technology (µBBB)

While parallel artificial membrane permeability assay (PAMPA) and cell-based transwell assays have been prevalent for the past two decades in CNS drug discovery, they suffer from being oversimplified and limited in terms of their biorelevancy, as in most cases they are a static two-compartment models, and for PAMPA, there is in fact no cell basis [169]. While well suited to the preliminary stages of R&D for small molecule screening with high throughput and low costs, for subsequent lead optimisation and testing, 3D models such as those offered by advances in microfluidics are warranted, particularly when they can be used synergistically with advanced cell culturing techniques, i.e., organoids. As has been demonstrated in several studies [170,171,172], near physiological shear stress can be incorporated into the model to better capture in vivo tightness and endothelial cell behaviour under the influence of simulated blood flow. This is of great importance for the toxicology studies of biocompatibility as nanomaterial safety must be demonstrable for species and disease-specific cumulative dosing studies of >1 month [64], which is not possible with transwell studies as the integrity of the endothelial cells cannot be maintained for long periods.

An ideal microfluidic platform for NTP testing requires integration of a number of key considerations. Firstly, the platform must be able to recapitulate the BBB endothelial cell vessel-like structures in 3D, mimic the cellular crosstalk and cross-regulation, simulate shear stress under flow and have a biocompatible basal membrane [173]. These requirements have been met to varying extents by modular configurations, ranging from the elementary sandwich design which evolved from transwell models, to parallel and 3D tubular networks, in addition to experimental inclusion of de novo microvessel formation over the use of microneedles. The article published by Oddo and colleagues [54] should be consulted for a more complete review of these designs, but, briefly, the drawbacks of the conventional sandwich configuration have been mitigated by more sophisticated designs, which will be considered here.

In general, the most successful models are comprised by co-culturing human or iPSC cells under constant perfusion in a glass synthetic microvasculature model [53], which has layered microchannels separated by microfabricated membranes with 3 µm gaps to generate patient and disease-specific models of the BBB, which can have a high degree of control and flexibility in terms of key parameters (Figure 6: such parameters include the tightness measured by TEER, permeability measured by small molecules such as dextran and inulin, tight junction protein expression measurable by microscopy and degree of shear stress). They can also be used to study the influence of pathogenesis, i.e., disease, such as AD/PD, ischaemic and hypoxic states, on the BBB behaviour and permeability of nanoparticles [174,175].

While adsorption and immobilisation of the endothelial cells to the glass layers of the model is generally achieved by silanisation or oxygen plasma activation as used in the model developed by Kim and colleagues [176], for longer term studies such as biocompatibility screening in cumulative dosing of NTPs, a covalent binding of the extracellular matrix is advantageous. Peng and colleagues [177] recently achieved a coated organ-in-chip using a photo cross-linkable copolymer that is amenable to in situ surface modification to model the contribution of the basal membrane to BBB formation and regulation, with high-throughput screening and simulated microchannel flow studies. The rate of medium flow through each chamber thus determines the shear stress, which can be regulated and modulated at will, which is much better for biorelevant studies than simple rotation devices and constrained geometries employed in dynamic transwell models.

Additionally, pumps, sensors such as electrical impendence sensing (EIS for nanotoxicity assessments) and electrodes can be readily employed in these models to measure key properties such as the TEER, pH and ionic concentration gradients and ensure constant monitoring as developed by Liang and colleagues [178]. As the nanoparticles will have a homogenous distribution through the medium and each compartment can be readily sampled and imaged with high resolution microscopy in real time without the requirement for labels and dyes, the effective delivered dose of nanoparticles and cumulative dose safety evaluations can be studied [64]. They are also a valuable tool as they will eliminate the influence of aggregation, gravity and buoyancy due to the laminar flow and artificial enhanced permeation and association of nanoparticles with the model itself as observed in transwell assays [52], thus giving more accurate and reliable data in relation to absolute nanoparticle delivery across the BBB.

While several papers give an excellent account of the advances in microfluidic models for disease modelling applications, this review is to give an account of the application of organ-on-chip models to the study of nanomaterials. For an in-depth account of such technologies for disease modelling, please refer to Holloway and colleagues [171] and Van der Helm and colleagues [179]. Briefly, however, a number of models given in these reviews have potential for developing nanoparticle treatments in specific disease states, such as in AD and glioblastoma multiforme (GBM).

While a number of organ-on-chip models have been used for in vitro drug development for evaluating efficacy and toxicity of novel drug compounds, such as the liver chip [180], kidney chip [181], gastrointestinal chip [182] and lung chip [183], the NVU/BBB chip is still relatively in its infancy, as it is arguably the most difficult to capture efficiently. Recent efforts have thus focused on developing more sophisticated models that employ synaptic activity in subcellular structures of the model, and optimising the models for specifically studying nanomaterial transport, as classically the models have been preferentially adopted for neuroscientific research applications. [184].

The search for biocompatible materials is also at a premium, as although that which has been used almost ubiquitously in the models published to date is utilitarian, the primary limitation is that it can absorb small organic compounds [185], which would seem to confound the findings of permeability studies for nanoparticles where it is used as the micro-fabrication material. Biofabrication has improved in recent years to improve the functional tightness of these chips (>2000 Ω cm^2^) [186], and indeed the constraints associated with employing two cell types only in a two compartment model has been recognised as being unsatisfactory. While Bang and colleagues [187] consolidated this latter issue and thus improved the models postulated by Booth and colleagues [188], and later Adriani and colleagues [189], to permit independent emulation of the internal and external vascular microenvironments, the use of rat-derived cells still implies cross-species translatability limitations.

Campisi and colleagues [190] thus realised a state-of-the-art model that combines the strengths of all of its forebearers, realising an iPSC-derived tri-culture model, which serves as a robust 3D platform for drug permeability studies. The perfusability and permeability were validated and comparable to other literature-derived data, demonstrating its potential for automated high-throughput drug transport studies. Using a coating such as that developed by Peng and colleagues [177], and culturing under constant flow to generate more BBB relevant microvascular formation and reduced permeability over a longer period, one could envisage a comprehensive BBB model for nanoparticle transport and safety studies.

Developing these concepts further, the studies of Caballero and colleagues [191], and more recently Vatine and colleagues [192], exemplify the potential of microfluidic platforms for personalised nanomedicine as cells obtained from affected patients can be used to create a disease-specific model of the BBB, with associated disruption of barrier integrity and downregulated transporter expression. As many studies have implicated receptor-mediated pathways of entry for nanoparticles, it is of pivotal importance to monitor how nanoparticles will behave with respect to an aberrant BBB microenvironment and how novel strategies such as that proposed by Bonakdar and colleagues [193] can be used to modulate the BBB with subsequent efficient delivery of nanoparticle therapeutics to the CNS.

A common feature of many CNS injuries and diseases is the induction of astrocyte reactivity leading to neuroinflammation [194], which has been accurately modelled and validated by Ahn and colleagues [195], which permits precision sampling and nanoparticle quantification for assaying transport and distribution in homeostatic and pathological states in their 3D organ-in-chip model. This could be integrated with other models to generate effective body-in-chip models, which would be invaluable for investigating cancer metastasis and neurodegenerative illnesses recognising the contribution of the brain–gut axis [172] and multiple system-mediated diseases such as hepatic encephalopathy and muscle lesion-induced CNS damage [56].

Thus, as alluded to by Wang and colleagues [56], for developing and testing small molecule delivery strategies for CNS disorders such as brain-targeted nanotheranostics, microfluidics will be imperative in clarifying such brain–organ intercommunication, and further to evaluate the efficacy and potential of therapeutic strategies such as those constituted by nanomedicines. As such, the trend towards personalised medicine and integrated in vitro models to reduce animal testing [170] will likely be resolved by iterations on the themes and recent trends in microfluidics, and constant improvements in technological capabilities and integrated understanding of the dynamic microenvironment of the BBB and how this nuanced complexity can be accurately represented in the lab, particularly with the advent of brain organoids developed by co-culture of iPSC and patient-derived cells, as discussed by Yu and colleagues [196] in a perspective article on this evolving field.

They note that while robust, many organoids have not achieved their full potential, and microfluidics may answer the current issues relating to the use of organoids as in vitro models. These include improving scale-up and size constraints for high-throughput drug screening; expediting the timescale for organoid formation, which is in the range of months at present; and recapitulating several key aspects, such as the contribution of microglia, shear stress and vascularisation to the integrated in vitro platforms and, moreover, simulating the dynamic nutrient, gas and waste exchange processes. Consolidating all of these aspects of organoid models will pave the way for generating predictive and biorelevant pharmacokinetic/pharmacodynamic profiles in reproducible and well-characterised models which are cost effective and commercially available to researchers in lieu of animal testing.

### 5.2. In Silico Simulated NTP Transport Studies

While the vast majority of contemporary studies have trended towards using in vitro microfluidics over transwell cell culture and more classical methods such PAMPA, for CNS-targeted nanoparticulate drugs, in silico screening strategies have also been relied upon [197]. While not as sensitive and translationally meaningful as in vitro studies due to most screening libraries lacking the necessary volume of data and permeability simplification being predicted based on algorithms and previous experimental data, machine learning and artificial intelligence mean that these methods have more predictive power when studies are carefully designed and optimised [60].

One of the fundamental properties of such studies is that the dataset selection employs information that is reproducible and orthogonal in nature, such that the cumulative findings have in vivo extrapolation significance. As Goodwin and colleagues [198] elucidate in their article, the most widely reported measure is by estimation of the logBB, which is analogous to the parameter measured in PAMPA, whereby passive diffusion is the assumed transport mechanism, with the ratio of solute concentration in plasma and brain in the two compartment simulation governing CNS penetrability. While most high-throughput software for nanomaterial studies in silico report this value, this largely proves inadequate for lead optimisation stages of drug development as it does not discriminate between free and plasma bound solute concentrations, does not map biodistribution and largely ignores the receptor-mediated mechanisms of transport that are pivotal for brain-targeting strategies for nanotheranostic platforms [199].

While models have become more sophisticated to include the contribution of endothelial surface area (Log PS) [200] and the CSF solute concentration (Log CSF) [201] to the quantitative structure activity relationships of nanomaterials using known physiochemical properties and biophysical descriptors, i.e., particle size, zeta potential, shape and Log P, etc., the most robust models are constituted by those that employ comprehensive molecular dynamics simulation (MDS) software packages [202]. These frequently employ machine learning and rule-based models of the BBB such as the modified Lipinski rule of five for predicting CNS penetration using regression analysis and established in vitro data [203].

Due to limitations implicit in such software in terms of the simplification of transport mechanism modelled, size of the computational cell and simulation time scales to model millions of nanoparticles interacting with an artificial lipid membrane, coarse grain models have been developed [204], which can be further modified to synthetically model disease states, such as PD, AD, MS and the highly heterogenous tumour microenvironment in GBM [205]. These models employ “pseudo atoms” to represent the nanoparticles with less degrees of freedom [204], to consolidate the study design and enable nanomaterial risk and nanosafety evaluations [205] in tandem with permeability studies to establish CNS activity. This is particularly imperative with the advent of elaborate nanotherapeutic engineering strategies, such as nanorobots [206], which are enhanced by optical or magnetic guided targeting to the brain as simulated by Pedram and colleagues [207].

As alluded to, all of these models require well-established and information-rich training sets, which have been bolstered by seminary efforts by teams such as Gao and colleagues [208]. For an in-depth review of recent computational molecular modelling strategies and the mathematical significance of the measured parameters in such algorithms, consult the recently published reviews by Shityakov and colleagues [206] and Kisala and colleagues [209]; a few essential features will be briefly discussed.

In silico models have an estimated 70% success rate in accurately predicting the Log BB [210], which is a pragmatic compromise given the constraints associated with modelling log PS and reproducibly generating such data across studies where different protocols, algorithms and regression analysis are employed. While discerning and classifying potential CNS compounds and nanoplatforms as CNS penetrating or non-penetrating (CNS^+^/CNS^−^) such as delphinidin-loaded nanoparticles for GBM, [205] hesperidin-loaded nanoparticles for carotid artery occlusion reperfusion [211] and cerium oxide nanoparticles for PD treatment [212], care must be exercised to ensure the measures are reliable.

The role of P-gp cannot be overlooked, as a CNS^+^ compound may be artificially deemed as a hit or lead candidate, when in fact it is rapidly metabolised or effluxed [213]. This is generally accounted for by the use of resampling and molecular docking simulations, which are simpler and faster than MDS. QSAR studies have thus benefitted from non-linear models, which have machine learning and resampling in-built, which has facilitated the advent of computational neural networks for high throughput nanoparticle permeability studies [214]. These models are more deterministic in nature for examining binding kinetics of nanoparticle–cell interactions, transport across the BBB and biodistribution/biofate. As the review of Singh and colleagues [206] highlights, the need for superior algorithms will only be met by interdisciplinary collaboration of computer scientists and researchers. The contribution of computational methods would arguably be two-fold in expediting the drug discovery process and, moreover, would aid in shifting the regulatory framework by rapid nanotoxicity evaluation and demonstration of biocompatibility [215]. Integrating in vitro and in silico methodologies to reduce animal testing would thus, as a prerequisite, require superior algorithms that incorporate active transport-mediated mechanisms in the model and facilitate generation of de novo nanotherapeutics with desirable BBB properties.

## 6. Rational Nanotheranostic Design for Accelerated In Vitro Testing

While all of the foregoing models have admissible limitations, a commonality in favour of their adoption of animal testing is that it controls more variables in the testing data. Notably, when studies include data obtained from commercially available models with characterised cell features such as morphology, confluency and increasingly using tissue of human origin in their construction, several degrees of freedom causing inconsistent data acquisition are removed. This in turn lends itself to allowing formulation scientists to rationally design the nanoformulation in a simplified and expedited manner, as the “trial and error” approach generally is necessitated by the fact that no two research groups can acquire congruent data, or indeed in many cases within two runs of the one experiment. When the models are commercially available, the nanoengineering can become the main focus, with versatile in vitro platforms that minimise the need for administration directly to animal subjects. This is an exciting prospect, particularly when one considers the laborious nature of acquiring and interpreting such data, and the resulting issues with up-scaling and reproducibility.

As the models recapitulate the most fundamentally important features of the human BBB, the attention can then be placed more on high-throughput screening and identifying novel strategies for crossing such a dynamic barrier, with additional NTP concentration measurements by direct sampling of various compartments of the model. This is particularly true of organ-on-chip models, which can simulate dynamic features such as shear stress and pathophysiological states such as hypoxia, lending freedom to the researcher in terms of manufacturing NTP platforms for testing irrespective of the intended specific indication. While the available nano-formulation strategies and designs have been extensively reviewed in other seminary papers, the most promising recent advances in nanotheranostic design and clinical candidates will be considered briefly here as they relate to in vitro testing.

### 6.1. Inorganic Nanotheranostic Clinical Pipeline

Despite the prevalence of the “valley of death” in nanotheranostic neuropharmaceutical development [215], a number of exceptional agents have demonstrated multifunctionality and versatility in indication with promising results. It is arguable that with more widespread adoption of in vitro modelling platforms for conducting transport and biocompatability studies, this number is set to increase [14]. At present, considerable precedence is given to their application in cancer nanotherapy [9,16] as this is arguably the most ubiquitous indication for most rationally designed nanoplatforms. A snapshot of the of privately and publicly funded clinical studies conducted around the world is given in Table 3 (www.clinicaltrials.gov, accessed on 19 September 2021).

In terms of rational design of nanotheranostic platforms, a number of key characteristics and design principles [16] should be considered from the outset in order to maximise the probabilistic outcomes of: (1) transport across the BBB, (2) biocompatibility and (3) measurable concentrations that are therapeutically useful. In essence, the fundamental goal for a formulation scientist is to design a nanotheranostic platform that uniformly crosses the BBB in a predictive and reproducible way, localises in the target regions and tissues of the brain or associated tumour/embolism and elicits a therapeutic effect (ideally with a sustained release profile) without causing significant accumulation or adverse effects.

Evidently, this is no trivial effort, particularly when considering the nuanced approach required for dose optimisation and balancing the diagnostic and therapeutic modalities in a homogenous and reproducible platform that is scalable to clinical settings [216]. As such, the in vitro modalities would frequently prove imperative to rational design as high-throughput and rapid screening of a platform can be conducted in inexpensive and simplified transwell models [217]. While limited in terms of ultimate predictive power, these would at least be robust enough to permit rapid indicative assays that are reproducible and a reliable indicator of probabilistic biocompatibility and BBB permeation.

### 6.2. On Current Engineering and Rational Design

Polymeric and metallic nanoplatforms constitute the most pervasive in the field for development of novel neuropharmaceuticals [16] due to the comparative lack of biomaterial-based studies. While numerous candidates, as alluded to earlier, have shown promise preclinically in vitro, and in some settings have reached phase 2 trials, the vast number of discontinued or unproven designs are a direct consequence of the “trial and error” philosophy [218], which has hampered research to date. Unfortunately, as a newly emerging field, the regulatory landscape is such that rigorous characterisation and validation of such elaborate nanoplatforms is mandated, yet clear guidance and approved specification suites are not available. Moreover, scalability and reproducibility concerns [219] are such that following unsatisfactory in vivo animal data, frequently the project is abandoned, or indeed the research team is forced to “start from scratch” and develop another platform which may meet a similar fate.

As such, what is urgently required is the ability to have rational design methodologies that not only improve hit probability in rapid in vitro screening in the models outlined in this review but also facilitate reconfiguration and modular construction which can circumvent commonly encountered troubleshooting in preclinical development. For instance, a commonality among the polymeric and metallic nanoparticulate systems such as dendrimers, gold and halfnium nanoparticles as well as SPIONS is that modification is limited to chemical alteration of monomers or post-processing of a polymer [16]. This not only lacks scope but can in numerous instances lead to compromises in terms of biocompatibility, drug encapsulation efficiency and prevention of premature release of either diagnostic or therapeutic modality.

Due to the foregoing, a novel promising nanoplatform has been developed, known as customisable telodendrimers [220]. While conventional polymerics are limited in scope due to heterogeneity due to radical or other polymerisation one-port reactions, these can be engineered by stepwise dendrimer block co-polymer synthesis and subsequent self-assembly with exquisite control of chemistries and reproducibility on scale-up. This is particularly useful in light of associated size-dependent limits (10 to 15 nm for AuNPs to permeate the BBB adequately [221]) for NTPs, shape-dependent permeation and the requirement for surface functionalisation and subsequent coating owing to the evolution of the protein corona [222] in addition to the highly anionic nature of the BBB endothelial cell surface.

In this regard, polyethylene glycol, when used as the hydrophilic moiety, would seem to confer advantages in facilitating customisable rational design, which can be actively targeted to the brain in stable platforms [16], which become increasingly important properties when it is observed that at such sizes, NTPs frequently accumulate in the liver and the brain in a seemingly non-saturable fashion when administered in naked form. While metallic NPs such as AGuIX have demonstrated good biocompatibility, accumulation on repeated administration within a chemotherapeutic regimen would be a cause of concern [223], particularly within the liver. Evidently, the ability to tailor the size and functionalisation in addition to external device-assisted delivery such as the aforementioned magnetic-guided therapy and focused ultrasound, would invariably result in preferential accumulation of therapeutically relevant concentrations of the NTPs across the BBB within the target tissues of the brain, which could be rapidly tested in various configurations and compositions in vitro to optimise the platform [224].

In their extensive nanoparticle engineering research efforts, Guo and colleagues [16] found that the best stability was obtained by using dendrimers composed of a low number of hydrophobic side chains, thus minimising the occurrence of aggregation and unfavourable geometric morphology. They have successfully developed chemical synthesis methods that facilitate the introduction of amphiphilic groups and reversible cross-linking and stimuli-responsive cleavage where appropriate [225]. These tuneable nanocarriers have numerous capabilities including enhanced encapsulation and drug loading efficiency, drug targeting and selective site-specific drug release (using pH or hypoxia responsive labile ester functionality). In particular, cholic acid, when introduced in G2 and G3 of a dendrimer, has resulted in the generation of robust micelles with self-assembly, biocompatibility and modular construction in that a wide array of drug binding moieties (DBMs) and diagnostic agents can be encapsulated in a single nanoplatform, with additional surface functionalisation with peptides for active targeting.

In the case of brain-specific delivery, one could envisage functionalisation with iRGD for active targeting, encapsulation of synergistic chemotherapeutic moieties (of both a hydrophobic and hydrophilic nature by virtue of functional segregated three-layer telodendrimers) and a contrast agent for enhanced in situ MRI for diagnosis, monitoring and follow-up. They have also successfully developed hybrid telodendrimers [226] that can facilitate in situ protein encapsulation for advanced delivery of protein therapeutics, without inducing the inherent immunogenicity or protein denaturation, which has hampered their conventional delivery. The binding affinity can be optimised based on the cargo protein characteristics rather than developing a nanoparticle and subsequently manipulating it to encapsulate a therapeutic protein.

In this way, in silico prediction and peptide chemistry can be fine tuned for optimised delivery and release. This could be further interfaced with rapid in vitro transwell assay to demonstrate permeability [227], and, due to their high stability and biocompatibility, it would be endeavoured that the nanosafety evaluation could be abbreviated, leading to more time being allotted to optimising the nanoplatform for maximal brain delivery. Furthermore, the high drug loading capacity and permissibility to encapsulate hydrophobic and hydrophilic chemotherapies in synergistic combinations [226] would arguably reduce the effective number of nanoparticles and thus the dose of the drug (or diagnostic contrast agent) that would have to be administered.

Exemplary efforts such as these would arguably be bolstered by the available in vitro models presented in this review for screening various formulations of such nanocarriers. In fact, there is ongoing research by Shi and colleagues [228] into the possibility to immobilise such telodendrimers in hydrogel resins for local controlled release depot formulations and for engineering protein-binding dendrons in size-exclusive resins as “nano-traps” for sepsis immune modulation, which could have potential applications in acute brain dysfunction immunomodulation associated with sepsis.

### 6.3. Unrealised Promise of Biomaterial-Inspired Rational NTP Design

Of the thus far explored nanotheranostics based on biomaterials, lipid-based systems are the most pervasive. However, in spite of clinically available liposomal preparations of chemotherapeutics such as doxorubicin (DOXIL), none have been approved to date for the treatment of CNS disorders [229], which is disappointing given the obviated advanatages of such nanocarriers that are capable of self-assembly and modularity with surfactant coatings to generate “niosomes” [18]. Dependent on their surface character (zeta potential), they can cross the BBB by receptor-mediated or adsorptive-mediated transport mechanisms [230]. As alluded to in previous sections, surface functionalisation with ligands such as CBSA, transferrin and glutathione or peptides such as RGD have generated potentially useful active-targeted nanocarriers for receptor-mediated transport across the BBB. However, in numerous instances, it is becoming increasingly clear that the advent of new technologies such as phage display [231] means that cell-derived nanoparticles have unrealised possibilities for advanced drug delivery to the CNS.

By directly leveraging the biomaterials on cell membranes, for example, the issues associated with traditional synthetic preparation methods and associated heterogeneity are largely circumvented [16]. Moreover, critical interactions such as the demonstrable deterministic nature of the evolution of the protein corona on administration [232] can be more precisely predicted and the biomimetic platform tailored as necessary. In one study conducted by Chen and colleagues, minor changes in lipid composition for formulating solid lipid nanoparticles (SLN) had a profound influence on surface charge properties and consequentially on biodistribution and tissue penetrance. They also determined that apoliprotein rather than vitronectin-rich corona optimises delivery to tumour cells for cancer nanotherapy, which is mirrored by other studies that demonstrated the potential use of apolipoprotein (apoE3) surface coating [233] for preferential delivery of porphrin-lipid nanotheranostics across the BBB.

In this regard, in vitro platforms will prove indispensable for studying such nuanced interactions, particularly where the cell models incorporate cells predominantly from human origin. The more information that can be garnered in terms of the evolution of the protein corona, corona–nanoplatform interactions and nanoplatform protein corona interactions with tumour cells and environments, the more customisable the platforms will become. There is endeavoured potential for patient-specific and tissue-specific delivery of biomimetic nanoplatforms [38] in the future by virtue of developing models based on extracted patient cells and media to tailor therapy and minimise adverse events. There is also scope for hybrid nanoparticles, in which an inorganic nanoparticle can be biosynthesised, leveraged for its unique properties and made more biomimetic by using cell-based coating and functionalisation [234].

Examples of this approach include red blood cell coating of nanoparticles [235], cancer cell membrane coating [236], bacterial cell membrane coating [237] and indeed macrophage-derived coatings [238]. The latter in particular are advantageous in that they facilitate enhanced stability, circulation times, reduced immunogenicity and sustained delivery like other cell-based delivery methods, but further augment delivery based on infiltration in response to inflammatory pathophysiology. The potential of such disease-dependent delivery across the BBB has been explored by various researchers for treating neuroinflammation following acute brain injury or sepsis, HIV [35], PD and various cancer indications [239] due to the enhanced penetrance and concentration of macrophages (with associated nanoparticles) in the tumour microenvironment. The in vitro models may prove fortuitous in studying macrophage interactions with drug cargo and associated issues relating to premature degradation by endolysin. They can also be utilised to test various formulations and loading strategies, as well as for demonstrating the maintenance of bioactivity following extraction and storage [240] as these are acknowledged issues limiting the scale-up and clinical translation of macrophage-mediated nanoparticle drug delivery.

## 7. Conclusions and Future Outlooks

While there have been notable advances in the treatment of CNS disorders due to increasing understanding of underlying pathophysiology, improved diagnostic capabilities and novel therapeutic strategies, the BBB remains a critical barrier to the treatment of a number of brain disorders, including classical neurodegenerative diseases such as AD, PD and MS, oncology, ischaemic stroke and traumatic brain injury. While nanotheranostics are an exciting prospect in regard to targeted and personalised therapy of therapeutic and imaging modalities that are otherwise impermeable or unacceptably toxic to the CNS, the clinical translation remains elusive with high attrition rates despite a number of promising preclinical candidates by industrious researchers.

One of the primary reasons for this is the need to better understand the intricate and often multi-faceted interactions of nanomaterials with biological systems highlighted by an increasing acknowledgement of the protein corona and the requirements of testing biocompatibility. While there is an increasing trend away from the dogmatic principles of relying on the EPR effect for nanomaterials and the properties that govern preferential entry across the BBB, elucidation of the various receptor-mediated and transport processes has not been entirely realised.

The other reason for a lack of translation is the issue with reproducibility and stability in relation to nanotheranostic design, synthesis, scale-up and testing, particularly in regard to characterising acceptable in vitro and in vivo models. For more predictive and meaningful studies to be conducted, coupled with an increasing desire to move away from in vivo testing with associated costs, ethical and logistical considerations along with inter-species differences potentially confounding the results of clinical data, more robust clinical models are mandated. As has been outlined in this review, no one true model can be employed to capture the dynamic and sophisticated nature of all aspects of the BBB, and so the selection of model for testing potential clinical nanotheranostics in novel drug delivery strategies requires a practical consideration of the aim of the study and the stage of research and development.

For lead identification, in silico screening followed by high-throughput screening in monolayer models seems the most feasible option to identify potential candidates with acceptable permeability characteristics. A co-culture or similar transwell model would then seem prudent for rapid biocompatibility assessments of nanotheranostic platforms consisting of different nanomaterials of both organic and synthetic origin, preferentially in human-derived or iPSC cell lines where possible. This will likely become a reality with increasing commercial availability of standardised and validated sources of immortalised or human-derived cells, or indeed the ready-to-use models themselves.

For lead optimisation and pre-clinical nanosafety evaluation and efficacy studies, microfluidic models including organ-on-chip and organoid models would seem to be the most acceptable choice. These lend themselves well to biorelevant studies that can capture key aspects of the BBB, including the key contribution of shear stress forces due to blood flow. Remarkably, advancements of such models have realised a recapitulation of not only the intact BBB but also of modelling the disruption of the BBB inherent to several disease states, and thus can rapidly and dynamically facilitate assessment of the potential of nanotheranostic platforms for acute management and therapeutic intervention in such conditions.

These are very welcome as a formulation strategy in light of the increased focus of the pharmaceutical industry in the use of biotechnological products such as antibodies, peptides and siRNA for the management of CNS disorders including glioblastoma multiforme, PD and AD. By their nature, such biopharmaceuticals require robust delivery vectors that can protect such delicate biopayloads and selectively deliver them to the target tissues of the brain across the BBB, prerequisites which seem to be met uniquely by nanoparticulate delivery systems.

Further still, it is endeavoured that brain-targeted nanotheranostic drug delivery systems can answer outstanding clinical questions for which there has not been a satisfactory answer to date, such as disappointments in the treatment of AD and PD, as well as the acute and chronic management of traumatic brain injury. It has been increasingly understood by research teams such as Piot-Grosjean and colleagues [240] that such insult to and disruption of the BBB imposes life threatening complications in the acute phase and often debilitating consequences for chronic cases with poor prognosis, particularly arising from physical sports. 

Unfortunately, as of yet, and despite promising efforts such as those outlined in this review, there are few candidate neuroprotectants and associated delivery strategies that proceed past the in vitro setting

The result for clinicians is a dearth of interventions for use in emergency situations to intervene at critical care points such as on admission to emergency departments for cases relating to traumatic brain injuries and ischaemia and the associated excitotoxicity and oxidative stress [45]. Understanding such pathophysiological process using sophisticated models such as organoids and microfluidics and the subsequent rational design of nanotherapeutic interventions could be life saving and also life changing for affected individuals. The use of validated biorelevant models of the BBB for drug discovery and formulation development will incontrovertibly improve both (1) the clinical pipeline affording a suite of treatments at the disposal of clinicians and (2) patient outcomes, which are the pre-eminent goal of any clinical research.

While targeted nanomaterial-mediated brain delivery across the BBB constitutes a promising direction for CNS disease treatment, to reach clinical significance, the consolidation of the seminary efforts by researchers in nanotechnological fields will hopefully be met by increasing availability and use of the appropriate robust modular in vitro technologies for nanoparticle testing reviewed here. By employing validated models at the pivotal stages of R&D to demonstrate prospective nanotheranostic biocompatibility and efficacy in vitro, increased success of clinical testing applications and reduced animal testing would be observed. Moreover, the neuropharmaceutical industry would burgeon with a witnessed resurgence of an expedited and cost-effective drug development pipeline. This would undeniably garner increased regulatory approvals of novel multifunctional therapeutics with the potential to revolutionise diagnostics and personalised therapeutics for neurological disorders in an ageing worldwide population.

Perhaps these models additionally offer a crucial beacon of hope in the contentious ongoing debate on the merit of “nanomedicine” in the context of advanced drug delivery. While the use of animal models has been the mainstay of clinical R&D, perhaps, as observed here, the expediency and efficiency of data collection generated from validated in vitro models can also serve to reduce and in the future potentially eliminate the requirement of laborious animal testing. Moreover, with exciting prospects in nano-engineering and self-assembly of rationally designed nanoplatforms, it would be hoped that abbreviated characterisation and troubleshooting would be observed. The direct result would be more time and resources being allocated by researchers to optimising the delivery across the BBB in preclinical testing to identify clinical candidates and a greater number of clinical trials to pave the way for nanomaterial-based CNS theranostics. The precedent set by such rationally designed therapies would constitute a basis upon which further agents could enter the pipeline and offset the appreciably high attrition rate and translational gaps observed in the field at the present time.

As exemplified by notable clinical candidates in the pipeline and approved nanomedical interventions to date, the novel phenomena that nanotheranostic interventions exhibit are not readily tested reproducibly in animals, and scale-up issues, as aforementioned, render the data garnered indeterminate in relation to potential human use. Rather, pilot studies in animals can be prioritised at early lead optimisation stages where warranted and justified, while in vitro and indeed in silico modelling approaches should become the obvious choice for expediting development and clinical testing.

Rather than observing this as an expensive and uncharted ground in drug research efforts, as presented, the increasing availability of validated and commercially available models should be viewed by researchers as an exciting next chapter in the ongoing search for cutting edge nanotechnology driven drug delivery systems in the wider context of personalised medicine. By reducing the number of animals being used in animal testing in favour of these in vitro approaches, a number of ethical, economical and data potentiality issues can be simultaneously consolidated generating platforms for nanotheranostic drug development with bolstered clinical potential for improving the treatment outcomes in CNS disorders. Solving the conundrum posed by crossing the robust BBB is best envisaged as a composite of a series of nano-discoveries using novel testing strategies outlined herein, rather than hoping for a macro-sized breakthrough garnered from repetitive and unreliable testing in the classical settings, which, to date, have not produced any promising results. To continue in this vein would seem to be a missed opportunity and in a sense potentially compromise the viability of nanomedical research, which would be an inexpiable development given the untold potential of such therapeutics and in light of the exemplary efforts of research teams worldwide for improving population quality of life and longevity.

## Figures and Tables

**Figure 1 nanomaterials-11-02632-f001:**
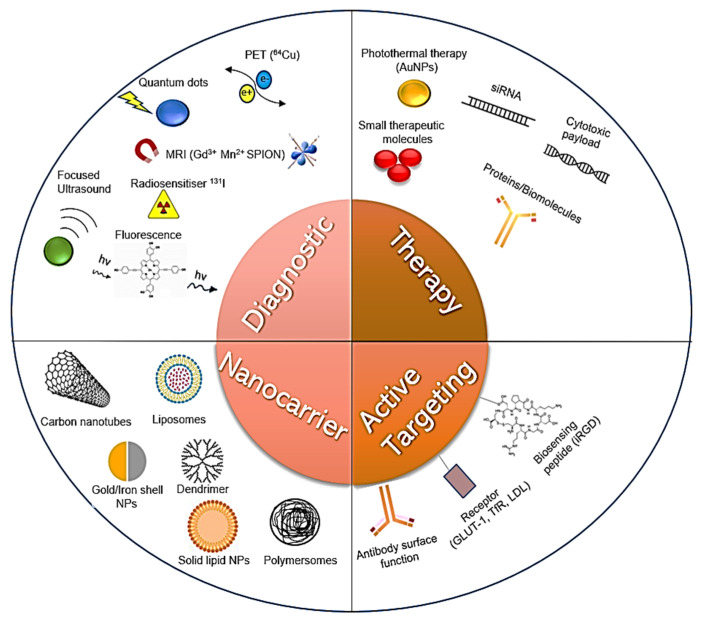
NTP platform design possibilities for targeted multifunctional imaging and treatment of CNS related disorders.

**Figure 2 nanomaterials-11-02632-f002:**
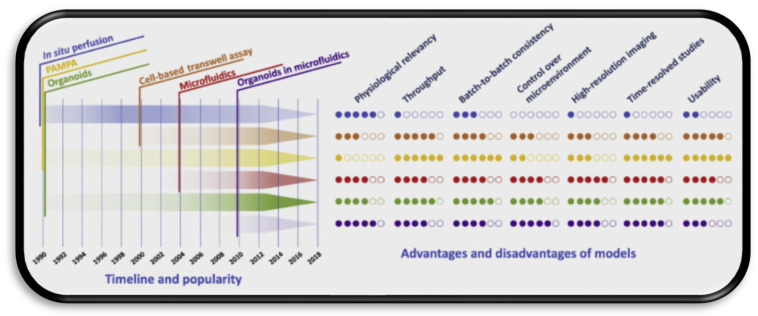
Trends in BBB models 1991–2018. The opacity of the lines in the graphic on left refer to overall popularity, and the shaded boxes on the right represent a qualitative 1–6 score where lower scores imply limitations and higher scores indicate relative strengths of a particular model. PAMPA = parallel artificial membrane permeability assay, a cell free assay used to screen the permeability of compounds based on the pass from a donor to acceptor compartment separated by an artificial lipid membrane. Reproduced with permission from Oddo and colleagues, Trends in biotechnology; published by Cell Press 2019 [54].

**Figure 3 nanomaterials-11-02632-f003:**
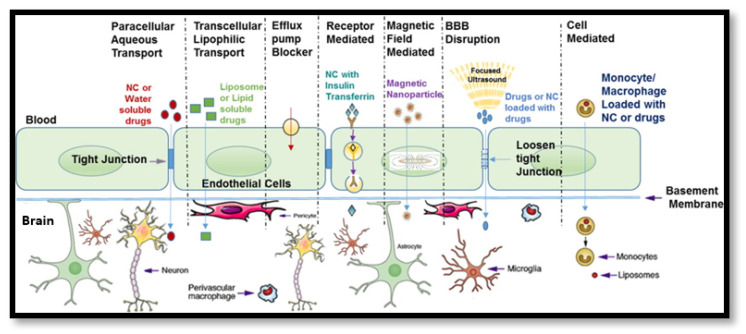
Summary of non-invasive transport mechanisms available for the delivery of nanoparticulate systems across the BBB. Reproduced from Nair and colleagues.

**Figure 4 nanomaterials-11-02632-f004:**
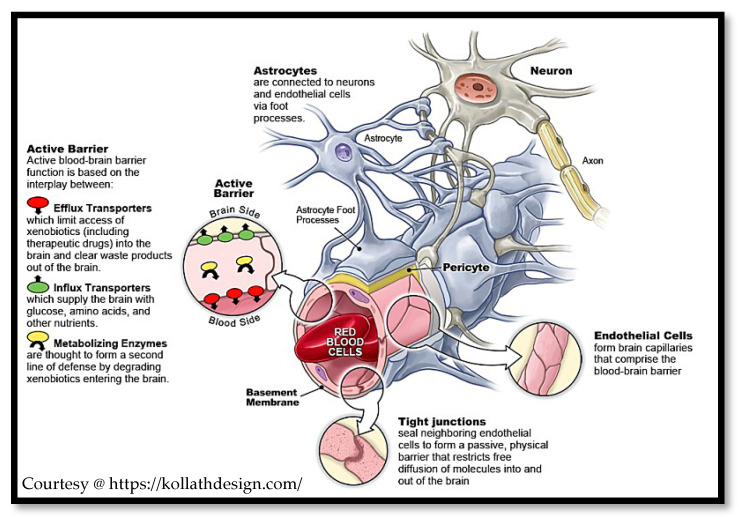
Replicating the dynamic barrier. A cross sectional view of the neurovascular unit that constitutes the BBB. Graphic courtesy of Mr. Richard Kollath (accessed on 29 April 2021).

**Figure 5 nanomaterials-11-02632-f005:**
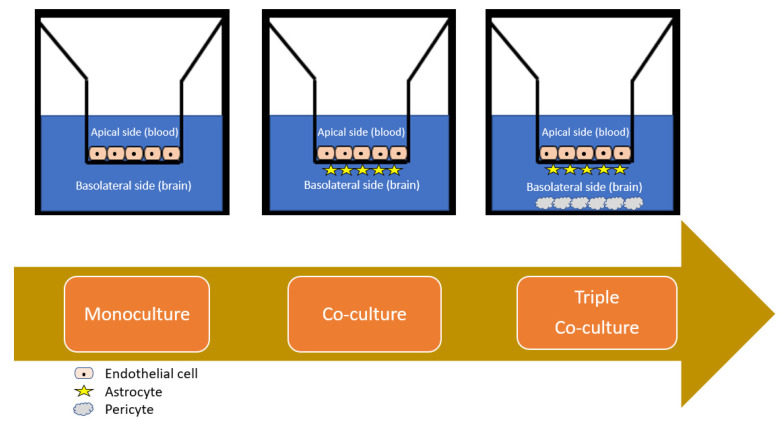
Co-culture models with increasing complexity and translational power from left to right. Note that the triple co-culture model comprises the essential neurovascular unit mimic with three cell types.

**Figure 6 nanomaterials-11-02632-f006:**
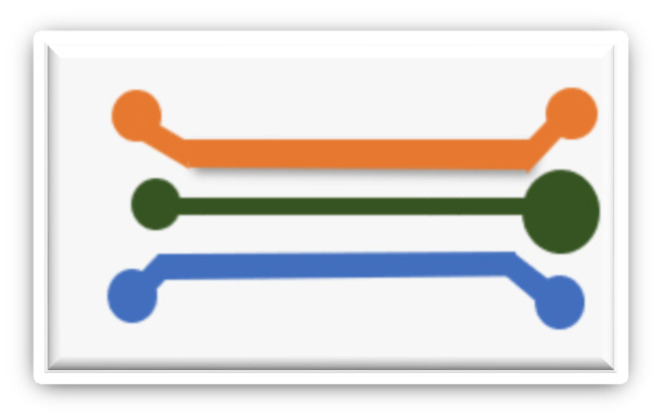
Top-down schematic of a typical microfluidic model with separated channels (approximately 2 cm in length. Bottom organ-on-chip model of BBB. Note each channel has microchannel separators marked by the colour interface of each channel (approximately 3 µm wide between blood and brain, and 50 µm between medium and brain)).

**Table 1 nanomaterials-11-02632-t001:** Main strategies for nanotheranostic drug delivery to the brain across the BBB.

Strategy	Benefits	Limitations
**BBB disruption by focused ultrasound**	Transient opening of the BBB facilitates increased concentration of NPs in brain	Inter-species limitations and variability of response between subjects’ limit findings [34]
**Magnetic field-guided delivery**	Enhanced imaging capabilities for diagnostics, in situ monitoring and follow-up of localisation and concentration and delivery guided by external device	Balance must be struck to attain efficient and specific hyperthermia while maintaining viability of healthy surrounding tissues in addition to observed development of thermotolerance in several subjects [62]
**Active transporter-mediated delivery**	Enhanced transport efficiency, active targeting and localisation of NPs administered intravenously	Homogenous surface functionalisation is difficult and requires additional characterisation, not applicable for larger NPs [64]
**Viral vectors**	Gene transfection efficiency high for delivery of siRNA and gene products	Safety concerns related to nature of delivery vector and dose optimisation issues in intravenous administration [65]
**Delivery via altered permeability due to pathophysiological state of BBB**	Improved probability of transport of NPs across the BBB due to leaky vasculature/altered endothelial cell morphology and confluency	Limited knowledge in relation to specific changes in the dynamic BBB environment in various brain disorders and pathophysiological states as well as dependence of response on disease model used limits predictive power [43]
**Cell-mediated delivery**	Ability to delivery NPs across exploiting natural products present in the body as a “Trojan horse”, thus improving circulation time, brain-targeting specificity and sustained delivery with reduced immunogenicity	Technical limitations pertaining to maintenance of viability during extraction, storage, formulation and administration and heterogeneity Incomplete characterisation of drug loading capacities and drug–macrophage interactions hampers clinical translation [35]
**Non-intravenous administration**	Can bypass the BBB by using alternative routes that are also less invasive e.g., nasal administration	Dose limitations and short residence time hamper nasal and pulmonary administration, in addition to propensity for localised irritation [18,66] Oral route largely precluded due to nature of NPs and addition of additional gastrointestinal barriers in addition to the BBB [67]

**Table 2 nanomaterials-11-02632-t002:** Executive summary of included cell-based models and associated validation markers.

Study	Model Type	Cell Line	Validation Markers
Chang 2009 [120]	Co-Culture	Bovine brain endothelial cells Rat mixed glial cells (60% astrocytes, 20% oligodendrocytes, and 20% microglia)	Monolayer integrity-Fluorescence staining Occludin tightness-Not explicitly stated but tight junction, LDL, TfR and y-glutanyl transpeptidase (y-GT) activity considered to be retained as per Cecchelli and colleagues 2007 [121] Permeability-Transferrin receptor inhibitor pre-treatment to demonstrate the specific TfR mediated endocytosis In vitro/in vivo correlation-Not explicitly reported but referenced as method comparable to that described in Dehouck and colleagues 1992 [122]
Georgieva 2011 [75]	Plasma membrane	Human brain endothelial cells [hCMEC/D3 cells]	Monolayer integrity-Fluorescence staining Occludin tightness-TEER (50 Ω cm^2^) Permeability-Hydrophilic tracers (sucrose/inulin) PECAM, ZO-I and MRP-I expression-Laser scanning confocal microscopy
Qiao 2012 [123]	Monolayer cell culture	Porcine brain endothelial cells	Monolayer integrity-Fluorescence staining Occludin tightness-TEER (700 Ω cm^2^) Permeability-Lactoferrin blocker pre-treatment to demonstrate Lf dependent transcytosis Iron delivery efficiency by Fe_3_O_4_ nanoparticles measured by graphite furnace atomic absorption spectrometry
Wagner 2012 [124]	Monolayer cell culture	Mouse brain endothelial cells (bEnd3 cells)	Monolayer integrity- Fluorescence staining Occludin tightness-Not explicitly stated, but it was determined that incubation with nanoparticles did not adversely affect tightness and integrity was retained Permeability-Receptor associated protein (RAP) blocking by co-incubation to demonstrate the LDL/LRPI dependent uptake mechanism LRPI, LDL and Apo-E receptor expression-Laser confocal scanning microscopy In vitro/in vivo correlation-TEM investigations of ApoE modified nanoparticles confirm endocytosis both in-vitro and in-vivo is mediated by the same pit forming endocytosis mechanism
Martins 2012 [125]	Monolayer cell culture Macrophage cell culture	Porcine brain endothelial cells Macrophage cell line (from frozen human plasma)	Monolayer integrity-Fluorescence staining Occludin tightness-Not explicitly stated but similar *in vitro* and *in vivo* data and low cytotoxicity infers representative of maintained integrity Permeability-Confocal fluorescence microscopy revealed higher uptake in endothelial cell culture model than macrophage model Biocompatibility-Alamar Blue cell viability assay (MIT) following solid lipid nanoparticle incubation
Gromnicova 2013 [126]	(I)Monolayer cell Culture (2)3D astrocyte Co- culture model	Human brain endothelial cells (I-BEC) Primary human astrocytes and brain endothelial cells (hCMEC/D3 cells)	Monolayer integrity-Fluorescence staining, not affected by incubation with glucose coated gold nanoparticles for 24 hOccludin tightness-Not explicitly stated but suc hco-culture models using human tissue are considered the most representative to simulate the in-vivo environment with hCMEC/D3 models of the BBB Permeability-Demonstrated by glucose coated nanoparticle transport across the model. with negligible diffusion or sedimentation which could confound findings oi static 2D/3D models. Trans-endothelial movement not Glut-I dependent but more probably size and charge dependent (favouring non-ionic character imparted by glucose coating of AuNPs) Biocompatibility-assay showed low cytotoxicity and low immunogenicity
Teow 2013 [127]	Monolayer cell culture	Human adenocarcinoma cell line (Caco-2) Porcine brain endothelial	Monolayer integrity-Inverted light microscopy Occludin tightness-TEER (800–1000 Ω cm^2^ for Caco-2 cells and 200–300 Ω cm^2^ for PBEC cells) removal of serum and addition of hydrocortisone improved tightness of the models Occludin and claudin expression-Not explicitly studied, but considered to be similar to described in Patabendige and colleagues 2012. P_app_ measurements of paclitaxel in both directions demonstrated the expression of p-gp in the monolayer models [128] Permeability—TEER measurements before and after experiments/incubation. Apparent permeability coefficient (Pan) was calculated from the equation P_app_ (cm/s) − (dQ/dt)/(C_o_xA)dQ/dt. which constitutes a robust quantitative value which facilitates orthogonal comparisons with other studies Biocompatibility-LDH assay showed low cytotoxicity of the dendrimer nanocarriers, and converse high cytotoxicity (antitumour activity) when conjugated with paclitaxel
Rempe 2014 [129]	Monolayer cell culture	Porcine brain endothelial cells	Monolayer integrity-Fluorescence staining and immunocytochemical analysis Occludin tightness-TEER measurements, although stated as percentages rather than absolute values Permeability-Hydrophilic tracers NC-sucrose and fluorescein isothiocyanate labelled bovine serum albumin (FITC-BSA). Found maximal permeability after four hours due to decrease in TEER and maximum values of Papp (cm/s) P-gp, occludin expression-Immunocytochemical analysis and implied from experimental data showing disruption of model integrity after four hours when incubated with the poly(cyanobutylacrylate) NPs, following by recovery of integrity to 80 % baseline TEER values Biocompatibility-Critical solids content of 26.62 µg/mL led to irreversible monolayer disruption, while those below half this value i.e., <13.31 µg/mL led to complete recovery of barrier integrity
Cramer 2014 [130]	Monolayer cell culture	Porcine brain endothelial cells Capillary choroid plexus cells	Occludin tightness-TEER, being expressed in percentages than absolute values Occludin expression-Western blot and immunochemistry Permeability-TEER measurements before and after treatment with AgNPs, confirmed by FITC-dextran P_app_ measurements, which were in agreement Biocompatibility-Neutral red uptake assay and microscopy to monitor cell morphology after incubation with AgNPs. The ethylene oxide nanoparticles were notably more cytotoxic than their citrate counterparts, with a critical concentration dependence (75 µg/mL) of monolayer disruption Pro-inflammatory capacity-Reactive oxygen nitrogen species, MMP-2 and COX-2 activity measured by zymography which was upregulated by ethylene oxide AgNPs but not for citrate AgNPs at standard concentrations (25 µg/mL)
**Bramini 2014** [131]	Monolayer cell culture	Human brain capillary microvascular endothelial cells (hCMEC/D3 cell line)	Monolayer integrity -Fluorescence staining Occludin tightness-TEER measurements and confocal microscopy, which found holes of total 200 µm^2^, and although these may have an exaggerating effect on the overall flux, they are accounted for in the mode. This would be consistent with those found in similar models, although this is frequently not investigated or reported Claudin expression Western blot and confocal microscopy Permeability-TEER measurements and fluorescent labelled permeability assay, Spinning disk confocal fluorescence microscopy and total internal reflection fluorescence microscopy (TIRFM) was used to quantify the translocation of the nanopartic1es in real time with 10 min exposure times of the carboxylated polystyrene NPs (40 nm and 100 nm sizes), demonstrating a preferential lysosomal accumulation within the model rather than true translocation
**Hanada 2014** [132]	Co-Culture	Rat brain microvascular endothelial cells Rat brain pericytes	Monolayer Integrity-Fluorescence staining Occludin tightness-TEER measurements before permeability measurements (150–300 Ω cm^2^) Perrneability—P_app_ (cm/s) of 30 nm, 100 nm, 400 nm silica nanoparticles compared With P_app_ of tracer sulforhodamine B. P_app_ studies of quantum dots with different surface charge functionalisations Biocompatability—Histological data confirm some degree of BBB disruption implied by thinning of the endothelial cell layers following hematoxylin and eosin (H&E) staining, though long term permeability assays indicated negligible adverse effects on BBB functionality In vitro/in vivo correlation-Not explicitly investigated but commercial BBB model used which has been previously validated by Nakagawa and colleagues using a suite of drug molecules including known substrates of MRP-I and p-gp [133]
**Shilo 2015** [134]	Monolayer cell culture	Mouse brain endothelial cells (bEnd3 cells)	Monolayer integrity-Fluorescence staining Occludin tightness-Most parameters were not explicitly investigated, but the bEnd3 monolayer is a validated and well established model, and imaging demonstrated it formed similarly to other studies Permeability-Flame atomic absorption spectrometry to quantify the AuNP uptake after 30 min incubation With various sizes of NPs, revealing preferential selection Of 70 nm barbiturate functionalized AuNPs for CT imaging applications (most total Au uptake), and 20 nm for drug loading (highest free surface area) In-vitro/ln-vivo correlation—Fluorescent confocal microscopy investigating interaction Of barbiturate loaded AuNPs with the model indicated specific pinocytosis mediated transport across the barrier, and some degree of association with the barrier itself
Xu 2015 [135]	Co-Culture	Rat microvascular endothelial cells Rat pericytes Rat astrocytes	Monolayer integrity—Fluorescence staining Occludin tightness-TEER (>200 Ω cm^2^) ZO-I, claudin 5 expression-Confocal microscopy Permeability-TEER measurements before and after incubation with AgNPs and polystyrene NPs as control, demonstrating BBB disruption by decreased resistance values after 24 h for the 10 µg/mL—AgNPs only (1 µg/mL AgNPs and control were unaffected) Biocompatibility—AgNPs at 10 µg/mL demonstrated reduced ZO-I expression, mitochondrial shrinkage, apoptosis and altered gene expression by immunostaining and microarray analysis of astrocytes *In vitro/ln vivo* correlation—Triple co-culture model gives high TJ protein expression and tightness for evaluating mechanisms of nanotoxicity and vasoactive compounds
**De Jong 2018** [136]	Filter free monolayer cell culture	Human microvascular brain endothelial cells (hCMEC/D3 cells)	Monolayer integrity—Fluorescence staining Occludin tightness—Not explicitly stated but model validated with permeability measurements ZO-I expression—Fluorescence microscopy Permeability—Model validated with P_app_ (cm/min) measurements for 4 kDa and 2000 kDa dextran, which were in agreement with 3D microfluidic organ on a chip models of the BBB. Also validated by collagenase A digestion of apical, cellular and basolateral fractions facilitating quantitative assessment of active LDL mediated transcytosis by fluorescence spectroscopy illustrating the quantitative mode of the model *In vitro/ln vivo* correlation—Filter-free model in a human cell line allowing quantitative and real time imaging of nanoparticle transport across the membrane
Zhang 2020 [137]	Transcellular monolayer cell culture	Mouse brain endothelial cells (bEnd3 cells)	Monolayer integrity—Fluorescence staining Occludin tightness—Not explicitly investigated, but permeability measurements used to validate the model and same protocols used as other studies which generated confluent monolayers with high TJ protein expression Permeability—P_app_ measurements of neutral nanoparticles used to validate the model quantitatively by fluorescence spectroscopy In vitro/in vivo correlation—Model mathematically expressed as a 2D barrier in terms of its bending rigidity, surface tension, viscoelasticity and surface charge, as well as ion concentration of the medium and size and charge properties of nanoparticles. Therefore, recapitulates several key aspects of electrochemical gradient driven endocytosis rather than receptor mediated targeting, which allows elucidation of key rational design properties for NP delivery to the brain
Sokolova 2020 [138]	Spheroid model	Human brain microvascular endothelial cells Human brain pericytes Human astrocytes Human microglia (iPSC derived) Human oligodendrocytes (iPSC derived) Human cortical neurons (iPSC derived)	Monolayer integrity-Fluorescence staining, confocal scanning microscopy Occludin tightness-Not explicitly investigated, but characterisation was conducted as per Zhou and colleagues who have extensively established and Validated this model (140] ZO-1, claudin-5, CD31, P-gp. GLUT-I expression—immunohistochemistry Biocompatibility-ATP production as a cell viability assay following incubation with dye (FAM-Alkyne) conjugated AuNPs for up to 24 h. showing negligible change demonstrating lack of clinically significant cytotoxicity *I**n vitro/**i**n vivo* correlation-3D Model employing six types of human or human related tissues which comprise the NVU, Hypoxia condition e.g. following ischaemic stroke recapitulated to determine influence of pathophysiology on nanoparticle behaviour and distribution
Kumarasamy 2021 [141]	Spheroid model	Human brain microvascular endothelial cells Human brain vascular pericytes Human astrocytes Rat neurons Rat microglia	Monolayer integrity-Fluorescence STEM Occludin tightness-Confocal laser scanning fluorescence microscopy, RNA-sequencing VE-cadherin, claudin-5, NG2 proteoglycan, GFAP β-III tubulin, iNOS, MAP-2 expression-Immunohistochemistry, Western Blot, SDS PAGE, ABC, GLUT 1,3,5, SLC, p-gp expression—RNA sequencing and PCA Permeability—FITC labelling and incubation Of nanoparticles with model for 24 h, followed by imaging Biocompatibility—Metabolic and morphological studies on endothelial and epithelial cells following incubation with different classes Of nanocarriers including graphene nanoplatesr carbon dots, polymeric and metallic nanoparticles In vitro/in vivo correlation-3D Model employing five cell types encompassing the BBB and associated microglia and neural networks with exquisite audio-visual data and extensive characterisation of an essentially ex-vivo NVU is the most biomimetic model type to date, though using rat and human derived cells can limit the translation and reproduction of results

**Table 3 nanomaterials-11-02632-t003:** Overview of current nanotheranostics in clinical development as of Q4 2021.

Product	Nanoplatform	Diagnostic Component	Therapeutic Component	Phase	Prospective Indication
AGuIX AGuIX	Polysiloxane matrix with gadolinium chelate radiosensitiser “Nano-Rad” Polysiloxane matrix with gadolinium chelate	Gadolinium Gadolinium	Gadolinium as radiotherapy adjuvant Temozolomide *	I Completed I/II Recruiting 2021	Whole brain radiation therapy in metastases Glioblastoma
Abraxane	Nano-albumin bound paclitaxel	In-situ MRI *	Paclitaxel, carboplatin and darvalumab *	II Recruiting 2017	Metastatic cancer of the head and neck
MM398	Nanoliposomal irinotecan	In-situ MRI * following Convection enhanced delivery	Irinotecan	I Active (Recruiting) 2013–2021	High grade Glioma
RXDX-107	Human serum albumin bound bendamustine	In-situ MRI *	Bendamustine (as dodecanol alkyl ester	I/1b (Terminated 2015)	Solid tumour (Locally advanced or metastatic)
SNB-101	SN-38 nanoparticles	In-situ MRI *	SN-38 (Active metabolite of irinotecan)	I Recruiting 2020	Metastatic head and neck cancer
NBTXR3 NBTXR3	Crystalline halfnium (HfO_2_) nanoparticles Crystalline halfnium (HfO_2_) nanoparticles	Halfnium nanoparticle as radiosensitiser Halfnium nanoparticle as radiosensitiser	Halfnium as radiotherapy adjuvant and anti—PD1 * (pembrolizumab) Halfnium as radiotherapy adjuvant and anti—PD1 * (pembrolizumab) with radiotherapy	I Recruiting 2018 II Recruiting 2021	Malignant high grade solid tumour Metastatic Head and neck cancer
Feraheme	Iron oxide nanoparticle with functionalised coating	Ferumoxytol as MRI enhancer	Macrophage polarisation by ferumoxytol *	I Recruiting 2017	Childhood brain neoplasm
Feraheme	Iron oxide nanoparticle with functionalised coating	Ferumoxytol as MRI enhancer Ferumoxytol for dynamic susceptibility contrast-enhanced MRI Ferumoxytol for neuroinflammatory imaging Ferumoxytol and gadolinium MRI	Magnetic guided Therapy * Gadolinium for dynamic contrast enhanced MRI/as radiosensitiser * Magnetic guided therapy * Magnetic guided therapy *	II Recruiting 2017 N/A 2018 I (Early) Completed II Recruiting 2021	Brain neoplasm Recurrent childhood brain neoplasm Primary brain neoplasm CNS degenerative disorder CNS infectious disorder CNS vascular malformation Haemorrhagic and ischaemic brain accident CNS neoplasm Cranial nerve disorder Metastatic malignant neoplasm in the brain
MM398 plus Feraheme	Nanoliposomal irinotecan and iron oxide nanoparticles	Ferumoxytol	Irinotecan	I Completed	Solid tumours Breast cancer wit hactive brain metastasis
CPC634	Polymeric micelles of docetaxel	In-Situ MRI *	Docetaxel	I Completed	Metastatic cancer Solid tumours

* Denotes implied or co-administered therapies or diagnostics that could be rationally designed as a consolidated nanoplatform.

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
