# Peer review of "Advances in Non-Animal Testing Approaches towards Accelerated Clinical Translation of Novel Nanotheranostic Therapeutics for Central Nervous System Disorders"

_nanomaterials, 2021, doi:10.3390/nano11102632_

Round 1

Reviewer 1 Report

The title of this review is:
"Advances in nanotheranostics to enable systemic, brain-targeted drug delivery"
The title does not reflect the content. As stated by the authors  " The specific interest of this paper is overcoming the BBB and how nanomaterials can be rationally designed and tested using modeling of the BBB to facilitate clinical translation of promising nanoplatforms, with abbreviated in vivo testing requirements ». 
Reading the review it appears clear that the review addresses many aspects related to BBB including in vitro models, etc…however, I regret that I could’t draw a clear conclusion after reading this paper on « how nanomaterials can be rationally designed ». Consequently, It would be nice to include 1 additional section in this review to summarize the most recent nanotheranostic efficient designs reported for crossing the BBB. Finally in the conclusion and outlooks, the authors should bring some remarks and concepts on how nanothernostics should be designed to cross BBB. 

Author Response

We thank the reviewer for these comments. Regarding the title, we agree with the reviewer and have therefore modified it to read "Advances in non-animal based testing approaches towards an accelerated clinical translation of novel nanotheranostic therapeutics for central nervous system disorders". We have also modified the abstract and manuscript to clarify the objectives and conclusions of our review article. Specifically, in the introduction, we have added new sections (lines 221-231, 258-263, 276-281) to explain further the objectives of our review article.

Following the reviewer's request, we have now added a new table which summarises the various strategies employed to cross the BBB using theranostic nanoparticles (line 507) and a new section entitled "Rational Nanotheranostic Design for Expedited In-vitro Testing" (from line 1337) which includes a new table summarising nanotheranostic systems in clinical trials (line 1375). Finally, we have added some clarifying statements in the "Conclusions" section.

Reviewer 2 Report

This review addresses in a clear and well-written way the diversity of targeting strategies for crossing the BBB, identifying the reasons for the lack of the clinical translation of research data generated so far, as well as the trends and future directions for in vitro BBB permeability testing. However, all these aspects are already known. My main concern about this paper is the superficial approach in what concerns the theranostics aspects. In my opinion, it lacks an in-depth intersection with theranostic nanosystems, in order to clearly differentiate from other review works. 

Author Response

We thank the reviewer for their feedback. Whilst we understand their point of view, we disagree regarding the main focus of this manuscript which is to give the reader an overview of the non-animal techniques available to test the crossing of the BBB by theranostic nanoparticles. Our view in this context, it is to explain the physiology of the BBB, the techniques available to test the passage of NTP through the BBB and summarise the TNPs available. This review does not aim to describe theranostic nanosystems in depth. However, in order to clarify the main intention of the review, we have now added new tables and sections (see details above) to elaborate on the topic under focus and thus give a clearer idea of the aim of the review. We hope that the changes made in this revised version will help convince the reviewer of the central aim of this work and the originality of our review article.

Reviewer 3 Report

The review manuscript was discussed about nanotheranostics for brain-targeted delivery. The authors suggested current stratus and limitations of the noninvasive approaches for BBB crossing especially focused on validated modeling systems to evaluate nanotheranostics efficacy in vitro to increase clinical success and reduce animal testing. The discussion is through and informative with many valuable references. If the authors can give figures or table to summarize the contents, for examples, current approaches for crossing BBB, this review would be much easier for readers to grab ideas.  

Author Response

We thank the reviewer for their positive comments. As suggested by the reviewer, we have now added two new tables to this revised manuscript describing the various strategies employed to cross the BBB (line 500) and international clinical trials with TNP (line 1375). We believe these additions have enhanced our manuscript and hope they adequately address the reviewer’s comment.

Round 2

Reviewer 1 Report

The authors did a great job in this revised version and responded to my comments, without no doubt this review is now ready to be accepted.